# On the Computational Limits of AI4S-RL: A Unified $\varepsilon-N$ Analysis

**Qili Shen**[1,2,*] **Kairui Feng**[1,2,*,†] **Ang He**[3], **Xuanhong Chen**[3,†] **Dake Zhang**[3,†]

[1]Tongji University, Shanghai, China
[2]Shanghai Innovation Institute, Shanghai, China
{2311788,kelvinfkr}@tongji.edu.cn

[3]Shanghai Jiao Tong University, Shanghai, China
angh@nvidia.com, {chenxuanhong,dk.zhang}@sjtu.edu.cn

## Abstract

Recent work increasingly adopts AI for Science (AI4S) models to replace expensive PDE solvers as simulation environments for reinforcement learning (RL), enabling faster training in complex physical control tasks. However, using approximate simulators introduces modeling errors that affect the learned policy. In this paper, we introduce a unified $\varepsilon$-$N$ framework that quantifies the minimal computational cost $N^*(\varepsilon)$ required for an AI4S model to ensure that tabular RL can estimate the value function with unbiasedness, with probability at least $1 - \delta$. This characterization allows us to connect surrogate accuracy, grid resolution, and RL policy quality under a shared probabilistic language. We analyze how the discretization level $K$ of AI4S and RL space governs both PDE surrogate error and RL lattice approximation error, and we employ spectral theory and Sobolev estimates to derive optimal grid strategies that minimize total cost while preserving learning fidelity. Our theory reveals that different systems – such as ODE- and PDE-governed environments – require different allocations of effort between physical simulation and RL optimization, which is consistent with the empirical results. Overall, our framework offers a principled foundation for designing efficient, scalable, and cost-aware AI4S-RL systems with provable learning guarantees.

## 1 Introduction

Reinforcement learning (RL) for PDE-constrained control promises transformative impact in domains such as fusion energy (Degrave et al., 2022) and climate modeling (Feng et al., 2025). Yet a fundamental obstacle remains: high-fidelity PDE simulations are far too costly to support the millions of interactions demanded by RL. A natural remedy is to replace PDEs with AI surrogates, but this introduces a critical and largely unexplored question: **how should the resolution of the surrogate be coordinated with the RL agent's discretization so as to minimize total computational cost while preserving policy accuracy?**

Classical RL theory assumes *stochastic* transition noise that vanishes under repeated sampling (Sutton et al., 1998; Azar et al., 2017), and robust RL extends this to worst-case distributional uncertainty (Derman et al., 2021; Agarwal & Zhang, 2022). In contrast, AI4S surrogates incur *deterministic* errors stemming from spatial discretization, temporal integration, and boundary approximation. Such errors do not average out with more data and can systematically bias policy learning. In a cart-pole system, if the surrogate discretization does not adequately capture the upright equilibrium, the learned controller will consistently overshoot; regardless of how fine the RL action resolution is, policy accuracy is ultimately constrained by the surrogate's resolution. Intuitively, minimizing regret under a fixed computational budget requires balancing the grid resolution of the AI4S surrogate with the decision resolution of the RL system.

---

*Equal contribution.
†Corresponding authors.

To address this trade-off, we introduce a framework that explicitly models discretization-induced bias. By sampling multiple trajectories from perturbed initial conditions within observation uncertainty bounds, we recast deterministic surrogate errors as a statistical inference problem. This structured sampling bridges numerical analysis in AI4S with RL learning theory, enabling principled resolution coordination between surrogates and agents. We focus on tabular RL to establish a necessary condition for learnability, providing a resolution lower bound that guarantees feasibility and guides hyperparameter design for deep RL in AI4S systems.

**Contributions.** We introduce the $\varepsilon$-$N$ framework that quantifies the minimal number of surrogate calls $N$ required to achieve $\varepsilon$-accurate value estimation with confidence $1 - \delta$, assuming optimal resolution allocation between AI4S simulation and RL exploration. Our analysis reveals that this optimization problem exhibits surprising complexity: optimal resolution ratios follow non-obvious fractional scaling laws (e.g., $\Delta t_r = (\Delta t_w)^{1/3}$ for tokamak control, where $\Delta t_r$ is the discrete timestep in the RL domain and $\Delta t_w$ is that in the AI4S/World domain), have exponential sensitivity to system dynamics, and vary significantly across different physical systems. We provide:

**1. Resolution coupling theory:** Our $\rho$–$K$ analysis shows that the forward prediction error $\rho$ scales as $\mathcal{O}(1/H_r + 1/K^d)$ in the high-resolution regime, where $H_r$ is the RL's temporal resolution (inverse decision frequency), $K$ is the ratio of physical grid cells to RL states, and $d$ is the spatial dimension of PDE system. This result quantifies how the resolutions of both the AI4S surrogate and the RL agent jointly determine the learning quality.

**2. Closed-form optimal allocations:** We derive system-specific expressions for the optimal resolution ratio $K^*$ that minimize computational cost. For instance, $K^* = (7/4 \cdot \exp(1/\lambda_1))^{1/3}$ for tokamak plasma control and $K^* = 2 \cdot \exp(1/\lambda_1)$ for cart-pole systems, where $\lambda_1$ denotes the dominant modal growth rate of the PDE dynamics.

**3. Empirical validation:** Experiments confirm that both tabular RL and DQN exhibit the predicted non-monotonic sensitivity to resolution, with off-optimal parameter choices requiring computational cost scaling as approximately $N^{1.6}$ relative to the optimal $N$ to reach the same accuracy. Moreover, there exists an optimal grid ratio $K$ and an optimal balance between $H_r$ and $H_w$, and these values are found to be relatively consistent across different learning algorithms.

**Related Literature.** Our work lies at the intersection of reinforcement learning theory and scientific computing. On the RL side, the tabular literature has established Probably Approximately Correct (PAC) bounds that characterize sample complexity (Sutton et al., 1998; Azar et al., 2017; Dann et al., 2017). Extensions to function approximation further reveal how this complexity depends on structural properties such as Bellman rank and eluder dimension (Jin et al., 2020; Wang et al., 2020; Jin et al., 2021). A key limitation of these analyses is their reliance on stochastic transition dynamics, where errors are assumed to average out. This assumption breaks down in scientific computing surrogates, where deterministic discretization errors introduce systematic, resolution-dependent biases. While robust RL frameworks (Derman et al., 2021; Agarwal & Zhang, 2022; Shi et al., 2023; Kwon et al., 2021) address model uncertainty through worst-case optimization, they primarily target irreducible stochastic noise. In contrast, our focus is on structured numerical biases, which we exploit to derive principled resolution–sample trade-offs.

On the scientific computing side, recent advances in neural solvers have accelerated PDE simulations by orders of magnitude. Neural operators (Li et al., 2020; Lu et al., 2021; Kovachki et al., 2023) and physics-informed neural networks (Raissi et al., 2019; Karniadakis et al., 2021) have enabled breakthroughs in weather prediction (Bi et al., 2023), turbulence modeling (Kochkov et al., 2021), and plasma control (Degrave et al., 2022). Although operator learning admits error bounds (Kovachki et al., 2023), prior work rarely investigates how surrogate resolution affects downstream RL performance. Classical PDE control methods rely on adjoints and Pontryagin's principle (Pontryagin, 2018), and recent applications of RL to PDEs (Farahmand et al., 2017; Han et al., 2018) typically assume access to true dynamics or ignore surrogate errors.

## 2 PRELIMINARIES

**Notation.** We study a hybrid system where an RL agent interacts with an AI4S surrogate approximating a PDE-governed environment. A comprehensive notation table is provided in Appendix A with key notations:

- Subscripts $r$ and $w$ denote RL and physical world parameters respectively.
- $K = \Delta x_r / \Delta x_w$ is the ratio of spatial resolutions between the RL and AI4S domains, where $\Delta x_r$ is the spatial grid size in the RL domain and $\Delta x_w$ is that in the AI4S domain.
- $H_r$ and $H_w$ denote the temporal discretization (time-step size) of the RL agent and the AI4S surrogate, respectively.
- $\varepsilon, \delta$ are the target accuracy and confidence levels.

**AI4S Surrogate Errors.** AI4S models approximate PDE solvers through neural operators. For a PDE solution operator $\mathcal{G} : f \mapsto u$ approximated by surrogate $\mathcal{G}_\theta$, the error decomposes as:

$$\|\mathcal{G}(f) - \mathcal{G}_\theta(f)\|_{L^2} \leq C_1 H^s + C_2 X^d \tag{2.1}$$

where $H$ is temporal discretization, $X$ is spatial resolution ratio, and $s, d$ are parameters determined by the dimension and property of the underlying PDE solver.

**Probably Approximately Correct (PAC) Learning in Tabular RL.** We define an episodic Markov Decision Process (MDP) as $\mathcal{M} = (\mathcal{S}, \mathcal{A}, P, R, H)$, where $\mathcal{S}$ is a finite state space with $|\mathcal{S}| = S$, $\mathcal{A}$ is a finite action space with $|\mathcal{A}| = A$, $P$ is an unknown transition kernel, $R : \mathcal{S} \times \mathcal{A} \to [0, 1]$ is the reward function, and $H$ is the episode horizon. For clarity, we omit the subscript $r$ here; in later sections, we will use $r$ to distinguish RL parameters from those of the underlying physical system.

We call that an algorithm is PAC-MDP if for any $\varepsilon, \delta \in (0, 1)$, with probability at least $1 - \delta$, it outputs a policy $\hat{\pi}$ satisfying $V_1^*(s_0) - V_1^{\hat{\pi}}(s_0) \leq \varepsilon$ after at most $\mathrm{poly}(S, A, H, 1/\varepsilon, \log(1/\delta))$ episodes. A classical result in tabular RL settings is that the UCB-VI algorithm Azar et al. (2017) achieves PAC-MDP guarantee with sample complexity $N_{\text{episodes}} = \widetilde{O}\left(\frac{SAH^3}{\varepsilon^2} \log \frac{1}{\delta}\right)$, where $\widetilde{O}$ hides logarithmic factors.

## 3 EMPIRICAL MOTIVATION: RESOLUTION COORDINATION IN AI4S-RL

Before developing our theoretical framework, we first empirically demonstrate that an optimal discretization scale exists in AI4S–RL systems. This balance between surrogate and agent resolutions follows a scaling relation across grids.

We utilize the simulated Cart-Pole environment, which is tractable for exhaustive search yet representative of continuous dynamical systems. We establish two distinct experimental settings to bridge theory and practice. First, we employ Tabular Value Iteration on the exact physics engine to serve as a theoretical oracle. Second, to strictly emulate the AI4S workflow where learned dynamics replace expensive solvers, we train Deep Q-Networks (DQN) within a surrogate environment governed by a pre-trained neural network. Across both settings, we vary the temporal resolution ratio $\log_{H_w} H_r$ and the spatial resolution ratio $K$, measuring the sample complexity required to achieve a value function error below 1%. Full experimental details are provided in Appendix C.5.3.

Figures 1 and 2 show that sample complexity does not improve monotonically with resolution. Instead, both methods exhibit similar optima: tabular RL at $(K = 1.5, \log_{H_w} H_r = 1/3)$, and Q-learning at $(K = 2.0, \log_{H_w} H_r = 1/2)$. Off-optimal parameter choices require computational cost scaling as approximately $N^{1.6}$ relative to the optimal $N$ in order to reach the same accuracy.

These findings highlight that conventional hyperparameter search strategies (e.g., grid or binary search) are ineffective in this setting: resolution parameters span multiple orders of magnitude, making brute-force exploration prohibitively expensive in high-dimensional PDE systems. This motivates the development of a theoretical framework that predicts optimal resolution trade-offs from system properties, rather than relying on empirical tuning.

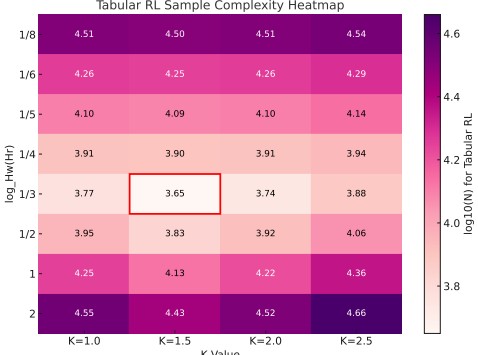 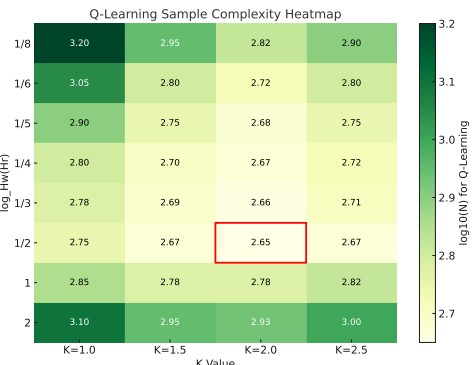

Figure 1: Sample complexity heatmap for the Cart-Pole system under tabular RL, showing non-monotonic dependence on resolution parameters. The red box indicates optimal configurations. Color intensity corresponds to $\log_{10}(N)$ as shown in the colorbar.

Figure 2: Sample complexity heatmap for the Cart-Pole system under Q-Learning, revealing algorithm-specific optimal configurations. Theoretical analysis is provided in Section 4.4.

# 4 ERROR PROPAGATION AND RESOLUTION COORDINATION IN AI4S-RL SYSTEMS

In this section, we develop a theoretical framework that predicts optimal resolution trade-offs from system properties. In Section 4.1, we show that the discretization of the AI4S surrogate itself determines whether the RL agent can converge to PAC guarantees. In Section 4.2, we analyze how the interaction between AI4S and RL discretization affects the convergence rate of the state transition matrix, using a magnetohydrodynamic (MHD) tokamak as a representative example. Building on these error models, Section 4.3 investigates the trade-off between the computational costs of surrogate construction and RL training. We derive the optimal resource allocation strategy—specifically the discretization resolutions—that minimizes the total cost required to achieve a PAC guarantee. Finally, in Section 4.4, we compute the optimal discretization and computational cost in four real-world systems to validate the framework.

## 4.1 SPECTRAL ANALYSIS OF ERROR AMPLIFICATION IN PDE-GOVERNED AI4S SYSTEMS

In AI4S systems governed by physical laws (e.g., PDEs), predictions are obtained by evolving the current state $y_0$ through a deterministic solver $f$ under action $a$, yielding $y_1 = f(y_0, a)$. However, $y_0$ is only observed up to grid size $\Delta y$, so the predicted next state lies in the range $[y_1^{\min}, y_1^{\max}]$, where the bounds are defined by $y_1^{\min} = \min_{\|\Delta y_0\| \leq \Delta y} f(y_0 + \Delta y_0, a)$ and $y_1^{\max} = \max_{\|\Delta y_0\| \leq \Delta y} f(y_0 + \Delta y_0, a)$. Here, $\Delta y_0$ is the measurement uncertainty in practice.

If the interval $[y_1^{\min}, y_1^{\max}]$ spans more than two full grid cells of PDE systems, the next state cannot be uniquely identified, and PAC guarantees are unattainable. In contrast, if the interval covers only a single full grid cell (with any additional coverage being partial), repeated sampling of the initial state allows us to estimate the next PDE state with arbitrarily high confidence. The convergence rate of this estimation is summarized in Theorem 1.

**Theorem 1** (Sample Complexity for $\delta$-Confidence Classification). *Consider repeated forward predictions from perturbed initial states, where the predicted frequency is the empirical probability of the next state falling into a given grid cell. Let $p$ denote the predicted frequency with which the true next state falls into the correct grid cell, and let $q = p_{\max}^{(j)}$ denote the maximum predicted frequency among all competing cells. To resolve the correct cell with confidence at least $1 - \delta$, the number of forward predictions required is bounded by* $n = \mathcal{O}\left( \dfrac{\log(1/\delta)}{\min_j \left( \Delta y^{(j)} - p_{\max}^{(j)} \right)^2} \right)$

*Proof in Appendix C.1.1.*

**Remark 1** (Time-Step Constraint for State Separability). *Consider the solution operator $f_t(y_0)$ of a nonlinear PDE. For observation perturbations $\eta$ with $\|\eta\| \leq \Delta y$, the linearized perturbation dynamics admit a modal decomposition $\eta(t) \sim \sum_k \hat{\eta}_k e^{\text{Re}(\lambda_k)t} \psi_k$, where $\{\lambda_k\}$ are the eigenvalues characterizing modal growth rates. Classification-cell separability requires that the dominant perturbation growth remains bounded, i.e., $e^{\lambda_1 \Delta t} \lesssim 1$, where $\lambda_1 := \max_k\{\text{Re}(\lambda_k)\}$. This implies a time-step constraint $\Delta t \lesssim 1/\lambda_1$.*

*Proof in Appendix C.1.2.*

From Theorem 1, when the two largest cell frequencies $\Delta y^{(j)}$ and $p_{\max}^{(j)}$ are close, where $(\Delta y^{(j)} - p_{\max}^{(j)})^2 \to 0$ and $n = \mathcal{O}\left(\frac{\log(1/\delta)}{(\Delta y^{(j)} - p_{\max}^{(j)})^2}\right) \to \infty$, PAC learning becomes unattainable. Remark 1 shows that smaller $\Delta t$ enlarges the frequency gap $(\Delta y^{(j)} - p_{\max}^{(j)})$, while if $\Delta t > 1/\lambda_1$, perturbations exceed $\Delta y$ and cell frequencies equalize, destroying state separability. Hence the constraint $\Delta t \lesssim 1/\lambda_1$ is necessary for PAC guarantees. We note that defining $\lambda_1$ as the global maximum is primarily for theoretical parsimony; utilizing a step-specific local growth rate in practice would not fundamentally alter the complexity analysis.

## 4.2 Error Coupling Between RL and PDE Spaces in AI4S Systems

In Section 4.1, we showed that the intrinsic dynamics and temporal discretization of the physical system determine whether PAC guarantees are achievable, and computed the sample complexity to reach PAC when possible.

In this section, we extend the analysis to mismatched discretizations between the RL and PDE spaces. Each RL action $a_r$ must be lifted to the PDE space for forward evolution, and the resulting physical state projected back for RL. These projections introduce errors, which may be amplified near PDE boundaries by nonlinear operators. Since RL actions often operate directly on boundaries (e.g., controlling droplet interfaces in tokamaks to avoid wall contact, or shaping fluid boundaries around an aircraft wing to increase lift), boundary accuracy is crucial. In MHD systems with droplet interfaces, the trace theorem implies that poor boundary approximation further enlarges these errors. To clarify the relationship between the two spaces, we define the discretization tuples $(H_r, S_r, A_r)$ for the RL agent and $(H_w, S_w, A_w)$ for the PDE system, where the components correspond to temporal, state, and action (or control parameter) granularities, respectively.

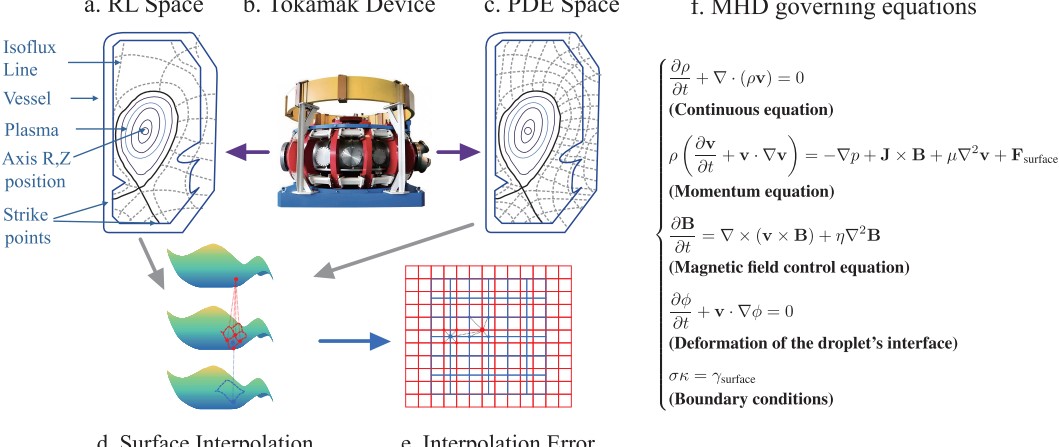

Figure 3: Illustration of Error Propagation in a Tokamak Machine RL controlling System.

**MHD System** As a representative case, in this section we focus on a magnetohydrodynamic (MHD) tokamak control task, where an RL agent manipulates boundary magnetic coils to suspend a plasma

droplet at the chamber center without wall contact Degrave et al. (2022); Ding et al. (2024). We assume observations of the MHD fluid and magnetic field on a predefined spatial lattice, and conduct numerical validation based on realistic tokamak configurations.

Figure 3 illustrates the governing PDEs, interface tracking methods, and control structure involved in the RL-based regulation of the droplet. The reward signal for reinforcement learning is defined as the minimum distance between the plasma droplet and the tokamak boundary, and the control actions correspond to adjustments of the magnetic field via coil currents. These equations define the error sources and discretization structure that form the basis for our multi-resolution analysis.

We assume continuous control over the magnetic field via time-dependent coil currents $\mathbf{I}_{\text{coil}}(t)$, enabling real-time adjustments to maintain droplet suspension without wall contact. Reinforcement learning algorithms adapt these control inputs based on observations of the droplet boundary and plasma dynamics. Key physical variables—including the magnetic field $\mathbf{B}$, current density $\mathbf{J}$, density $\rho$, velocity field $\mathbf{v}$, and interface function $\phi$—are continuously monitored and regulated to optimize positioning and stability.

**Interface Error Propagation via Control-Boundary Coupling**   In the tokamak control setting described above, RL actions do not directly affect the plasma interface but propagate indirectly from boundary coils through multiple physical fields. This indirect coupling introduces compounding discretization errors. We first analyze single-equation timestep errors, then demonstrate their coupling across the full system.

The control-to-interface coupling follows a hierarchical propagation path:

$$\text{RL action } a_r \xrightarrow{\text{projection}} \Delta a_w \xrightarrow{\text{boundary}} \mathbf{B}|_{\partial\Omega} \xrightarrow{\text{curl}} \mathbf{J} \xrightarrow{\text{Lorentz}} \mathbf{v} \xrightarrow{\text{advection}} \phi \tag{4.1}$$

where each arrow represents a potential error amplification point. The magnetic field $\mathbf{B}$ controlled at the boundary generates current density $\mathbf{J} = \nabla \times \mathbf{B}$, which produces Lorentz forces that drive the velocity field $\mathbf{v}$, ultimately advecting the interface level set $\phi$.

At each stage, discretization errors accumulate. The RL action $a_r$ must be mapped to physical control parameters $a_w$, introducing projection error $\mathcal{O}(\Delta a_r - \Delta a_w)$. Magnetic field boundary conditions suffer from reduced regularity, yielding $\|\mathbf{B} - \mathbf{B}_h\|_{L^2(\partial\Omega)} = \mathcal{O}(\|\Delta\mathbf{x}_{p,\text{bd}}\|^{1/2})$ by the trace theorem. These control errors then propagate nonlinearly into the velocity field as $\Delta\mathbf{v} = \mathcal{O}(\Delta a_w^{1/2})$, affecting the entire flow dynamics.

The droplet interface evolves according to the level set equation $\frac{\partial\phi}{\partial t} + \mathbf{v} \cdot \nabla\phi = 0$. Under forward Euler time discretization and central spatial differences, the local truncation error per PDE timestep $k$ becomes $\tau^{(k)} = \mathcal{O}(\Delta t_w) + \mathcal{O}(\|\Delta\mathbf{x}_{p,\text{int}}\|)$.

However, incorporating the velocity perturbations from boundary control and the reduced regularity at interfaces, the complete per-step interface error is:

$$\Delta_\phi^{(k)} = \underbrace{\mathcal{O}(\Delta t_w)}_{\text{temporal}} + \underbrace{\mathcal{O}(\|\Delta\mathbf{x}_{p,\text{int}}\|)}_{\text{spatial interior}} + \underbrace{\mathcal{O}(\Delta a_w^{1/2})}_{\text{control}} + \underbrace{\mathcal{O}(\|\Delta\mathbf{x}_{p,\text{bd}}\|^{1/2})}_{\text{boundary}} + \underbrace{\mathcal{O}(\Delta a_w^{1/2} \cdot \|\Delta\mathbf{x}_{p,\text{bd}}\|^{1/2})}_{\text{coupled}}$$
$$\tag{4.2}$$

In the following paragraph, we will couple this error with contributions from other governing equations to enable a comprehensive analysis.

**Refined Total Error Decomposition with Time-Scale Separation**   By extending the interface error structure discussed previously, we generalize the analysis to a broader class of PDE-based surrogate environments (details in Supplementary Information). Over a single RL step, the total prediction error can be decomposed as $\Delta_{\text{total}} = C_1\|\Delta\mathbf{x}_{r,\text{int}}\| + C_2\|\Delta\mathbf{x}_{r,\text{bd}}\|^{1/2} + C_3\Delta a_r^{1/2} + C_4\Delta t_r + C_5\frac{\Delta t_r}{\Delta t_w}(\|\Delta\mathbf{x}_{p,\text{int}}\| + \Delta a_w^{1/2} + \|\Delta\mathbf{x}_{p,\text{bd}}\|^{1/2} + \Delta a_w^{1/2}\|\Delta\mathbf{x}_{p,\text{bd}}\|^{1/2})$, where the first four terms represent RL-side errors (interior space, boundary space, action space, and temporal discretization), and the last term captures PDE surrogate errors amplified over the RL-to-physics time-scale ratio $\Delta t_r/\Delta t_w$.

To ensure that the RL transition kernel remains distinguishable under observation uncertainty $\Delta y$, we impose the condition $\Delta_{\text{total}} = \mathcal{O}(\Delta y)$. This leads to the following resolution matching constraints

between RL and AI4S components:

$$\|\Delta\mathbf{x}_{r,\text{int}}\| \sim \|\Delta\mathbf{x}_{p,\text{int}}\| \sim \Delta a_r^{1/2} \sim \Delta a_w^{1/2} \sim \Delta t_r \sim \Delta y,$$
$$\|\Delta\mathbf{x}_{r,\text{bd}}\| \sim \|\Delta\mathbf{x}_{p,\text{bd}}\| \sim \Delta y^2, \quad \Delta t_w \ll \Delta y. \tag{4.3}$$

This decomposition makes explicit how discretization scales in the RL and AI4S pipelines must be jointly selected to ensure unbiased value estimation. In particular, the term $\Delta t_r / \Delta t_w$ highlights how fine-scale physical errors can be amplified over coarser RL horizons, motivating resolution-aware control design.

In Section 4.1, we analyzed how to approximate the RL transition kernel under observation noise $\Delta y$. Here, we incorporate numerical errors from the AI4S prediction and define $\rho$ as the misclassification rate after one RL–to–AI4S projection and back. Concretely, let $p$ denote the probability that the true next physical state given one observed state falls into the correct grid cell after (i) projecting the observed PDE state into the RL grid, (ii) selecting the truncated optimal RL action, and (iii) mapping this action back to the PDE. Then $\rho := 1 - p$, which quantifies the probability that one RL–PDE interaction fails to preserve grid-level consistency.

From Eq. 4.3, total numerical error $\Delta_{\text{total}} \sim \frac{C_1}{K^3}\Delta y \sim \frac{C_1}{K^3}\Delta t_r$, and letting the RL temporal resolution be $H_r = 1/\Delta t_r$ (we also use this further on as a unit for calculating total computational cost), the total error rate is:

$$\rho = 1 - \frac{\Delta y}{\lambda_1 \Delta y / H_r + \Delta y + \Delta_{total}} = 1 - \frac{1}{\lambda_1/H_r + 1 + C_1/K^3}. \tag{4.4}$$

Hence, in the high-resolution limit (when $H_r$ and $K^3$ are large) $\rho = \mathcal{O}\left(\frac{1}{H_r} + \frac{1}{K^3}\right)$, showing that finer RL temporal resolution and AI4S spatial refinement both reduce the forward projection error rate at inverse polynomial rates.

From more general analysis on PDE scales, we could get:

**Theorem 2** ($\rho$-$K$ Analysis for $d$-Dimensional Systems). *For a $d$-dimensional PDE system, the forward projection error rate is: $\rho = 1 - \frac{1}{\lambda_1/H_r + 1 + C_1/K^d}$, where the numerical error scales as $\Delta_{total} \sim C_1 K^{-d}\Delta y$ from spatial discretization. In the high-resolution limit: $\rho = \mathcal{O}\left(\frac{1}{H_r} + \frac{1}{K^d}\right)$.*

*Proof in Appendix C.1.4.*

## 4.3 OPTIMAL COMPUTATIONAL COST ALLOCATION BETWEEN RL AND AI4S

To translate these PAC-possible AI4S-RL analysis into practical system design guidance, this section establishes a computational resource allocation framework for AI4S-RL systems. Since AI4S models are generally trained under fixed computational budgets, we aim to derive how their discretization parameters—denoted as $H_w, S_w, A_w$—can be aligned with the RL-side parameters $H_r, S_r, A_r$ to ensure computational and statistical consistency.

On the physical side, computational cost scales as $H_w \cdot S_w \cdot A_w$. For tabular RL, classical UCB-VI methods require a sample complexity of $\mathcal{O}(H_r^4 S_r A_r)$ for value function convergence without any assumptions on transition smoothness. Based on our transition identifiability analysis, an improved bound of $\mathcal{O}(H_r S_r A_r \cdot \frac{\log(1/\delta)}{\min_j\left(\Delta y^{(j)} - p_{\max}^{(j)}\right)^2})$ is achievable when the transition matrix is known up to statistical confidence $\delta$, reducing the RL problem to dynamic programming. From our $\rho$-$K$ analysis, we approximate $p_{\max}^{(j)} \lesssim \rho\Delta y \sim \frac{1}{H_r}\Delta y$, leading to a refined sample complexity of $\mathcal{O}\left(H_r^3 S_r A_r \cdot \log(1/\delta)\right)$, which we adopt in the remainder of this section.

We then align RL and AI4S computational costs via the relation $H_w S_w A_w = H_r^3 S_r A_r$. From the interface and grid-based error analysis in Section 4.2 for MHD system, and the temporal resolution bound in Section 4.1, we discretize the physical space in three spatial dimensions $(x, y, z)$, where: $A_r = \mathcal{O}(A_w) = \mathcal{O}(x_w^2)$, $S_r = \mathcal{O}(S_w) = \mathcal{O}(x_w^6)$, $H_r = \mathcal{O}(H_w^{1/3})$, $H_w = \mathcal{O}(x_w) \lesssim \frac{1}{\lambda_1}$. This constraint also respects CFL stability conditions for fluid systems.

By substituting these scaling relations into our cost-balancing equation, we can express the total computational cost as a function of the resolution ratio $K$, leading to the following theorem.

**Theorem 3** (Optimal Resolution with System-Dependent Scaling). *For a physical system subject to projection error with state space scaling $S_r \sim K^\alpha$ and action space scaling $A_r \sim K^\beta$, the computational balance condition $H_r^3 S_r A_r \sim H_w S_w A_w$ ensures equivalence between the RL surrogate resolution and the underlying PDE world resolution. Minimizing the overall computational costs,*

$$Cost(K) = H_r^3 K^{\alpha+\beta} \cdot \left( \frac{\log(1/\delta)}{\varepsilon^2} \right) \cdot \left( \frac{1}{1 - \frac{1}{H_r} - \frac{1}{K^d}} \right)^2 \tag{4.5}$$

*yields optimal resolution ratio between RL and AI4S space:*

$$K^* = \left( \frac{\alpha + \beta + 2d}{(\alpha+\beta)(1 - H_r^{-1})} \right)^{1/d} \approx \left( \frac{\alpha + \beta + 2d}{\alpha + \beta} \right)^{1/d} \cdot \exp\left( \frac{1}{d\lambda_1} \right) \tag{4.6}$$

*when $H_r \gtrsim \lambda_1 \gg 1$.*

*Proof in Appendix C.1.5.*

**Remark 2** (Impact of Actuation Topology). *The scaling exponents derive from the Trace Theorem based on the control location.* **Boundary-actuated systems** *(e.g., tokamak) require quadratic refinement due to boundary regularity loss, yielding $\alpha = 2d$ and $\beta = 2d_a$. Conversely,* **interior-actuated systems** *(e.g., heat sequencing) follow standard volumetric scaling with $\alpha = d$ and $\beta = d_a$. where $d$ denotes the physical spatial dimension and $d_a$ the action dimension. This distinction dictates the optimal resolution $K^*$ and the cost disparities shown in Table 1.*

## 4.4 THEORETICAL ANALYSIS ACROSS PHYSICAL SYSTEMS

Building on our theoretical framework, we now evaluate how surrogate discretization, spectral response, and RL precision jointly affect sample complexity across four representative AI4S-RL systems: (1) Tokamak plasma control, (2) Turbulent airfoil regulation, (3) Teppanyaki heat sequencing, and (4) Cart-pole stabilization (Details in Supplementary). Figure 4 illustrates their discretized state spaces.

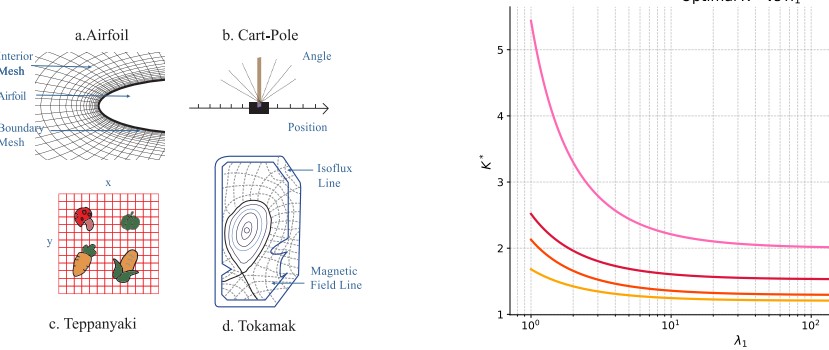

Figure 4: State space discretization for four AI4S-RL systems.

Figure 5: Log-scale plot of $K^*$ vs. modal growth $\lambda_1$.

We adopted an $\varepsilon-N$ perspective: to ensure value function estimation within error $\varepsilon$ and confidence $1 - \delta$, what is the minimal computational cost $N$, given surrogate discretization and physical uncertainty? Based on our analysis, we can now systematically achieve optimal resolution allocation under $(\varepsilon, \delta)$ accuracy constraints through a unified framework that integrates error normalization, identifiability analysis, and closed-form cost expressions—thereby minimizing the total computational cost of AI4S-RL systems while ensuring reliable policy estimation.

Table 1 summarizes scaling relations and optimal resolution ratios. Systems with strong boundary observability (e.g., tokamak, airfoil) require quadratic scaling in state-action discretization relative

to mesh resolution, while low-dimensional systems tolerate aggressive upsampling without cost explosion. Figure 5 shows how the optimal $K^*$ saturates with increasing $\lambda_1$, especially in high-dimensional PDEs. In contrast, ODE-based systems benefit more directly from higher surrogate resolution.

Our theoretical framework provides insights into the empirical results from Figures 1 and 2. For tabular RL, the observed optimum at $(K = 1.5, \log_{H_w} H_r = 1/3)$ requires $10^{3.65} \approx 4500$ samples, while the worst configuration at $(K = 2.5, \log_{H_w} H_r = 2)$ needs $10^{4.66} \approx 46000$ samples—a 10-fold increase. This translates to computational cost scaling approximately as $N^{1.6}$ when deviating from optimal parameters, consistent with our claim in the abstract. Q-learning achieves its minimum at $(K = 2.0, \log_{H_w} H_r = 1/2)$ with only $10^{2.65} \approx 450$ samples, demonstrating that function approximation can effectively exploit state similarity to reduce sample complexity by an order of magnitude.

Table 1: Optimal Resolution Matching and Computation Cost Across Systems under $\varepsilon$ error of value function estimation and $1 - \delta$ confidence level

| System | Resolution Scaling | Computational Cost | Optimal $K^*$ |
|---|---|---|---|
| Tokamak Control | $A_r = \mathcal{O}(A_w) = \mathcal{O}(x_w^2)$ 
 $S_r = \mathcal{O}(S_w) = \mathcal{O}(x_w^6)$ 
 $H_r = H_w^{1/3} \cdot \mathcal{O}(x_w) < 1/\lambda_1$ | $\dfrac{H_r^9 K^8}{\varepsilon^2} \cdot \dfrac{\log(1/\delta)}{(1 - \frac{1}{H_r} - \frac{1}{K^3})^2}$ | $\left(\dfrac{7}{4}\right)^{1/3} \cdot \exp\left(\dfrac{1}{3\lambda_1}\right)$ |
| Airfoil Control | $A_r = \mathcal{O}(A_w) = \mathcal{O}(x_w^2)$ 
 $S_r = \mathcal{O}(S_w) = \mathcal{O}(x_w^4)$ 
 $H_r = H_w^{1/3} \cdot \mathcal{O}(x_w) < 1/\lambda_1$ | $\dfrac{H_r^7 K^6}{\varepsilon^2} \cdot \dfrac{\log(1/\delta)}{(1 - \frac{1}{H_r} - \frac{1}{K^2})^2}$ | $\left(\dfrac{5}{3}\right)^{1/2} \cdot \exp\left(\dfrac{1}{2\lambda_1}\right)$ |
| Teppanyaki Plate | $A_r = \mathcal{O}(A_w) = \mathcal{O}(x_w)$ 
 $S_r = \mathcal{O}(S_w) = \mathcal{O}(x_w^2)$ 
 $H_r = H_w^{1/3} \cdot \mathcal{O}(x_w) < 1/\lambda_1$ | $\dfrac{H_r^4 K^3}{\varepsilon^2} \cdot \dfrac{\log(1/\delta)}{(1 - \frac{1}{H_r} - \frac{1}{K^2})^2}$ | $\left(\dfrac{7}{3}\right)^{1/2} \cdot \exp\left(\dfrac{1}{2\lambda_1}\right)$ |
| Cart-Pole System | $A_r = \mathcal{O}(A_w) = \mathcal{O}(x_w)$ 
 $S_r = \mathcal{O}(S_w) = \mathcal{O}(x_w)$ 
 $H_r = H_w^{1/3} \cdot \mathcal{O}(x_w) < 1/\lambda_1$ | $\dfrac{H_r^2 K^2}{\varepsilon^2} \cdot \dfrac{\log(1/\delta)}{(1 - \frac{1}{H_r} - \frac{1}{K})^2}$ | $2 \cdot \exp\left(\dfrac{1}{\lambda_1}\right)$ |

These results suggest that systems with high spectral amplification yield diminishing returns from AI4S refinement unless RL granularity is co-optimized. Our $\varepsilon$–$N$ framework thus provides a principled tool for balancing simulation fidelity and RL efficiency under finite compute budgets.

## 4.5 EMPIRICAL VALIDATION ON PDE-GOVERNED CONTROL: TEPPANYAKI HEAT SEQUENCING

While the Cart-Pole experiments in Section 3 successfully demonstrate resolution trade-offs, they rely on a low-dimensional ODE system with learned dynamics. To rigorously validate our theoretical predictions in high-dimensional PDE environments with continuous state-action spaces, we conduct a comprehensive study on a two-dimensional control problem: multi-item heat sequencing on a teppanyaki cooking surface. This environment is governed by the canonical two-dimensional heat diffusion equation $\partial T/\partial t = \alpha \nabla^2 T + Q(\mathbf{x}, t)$, representing a broad class of parabolic PDEs encountered in AI4S applications (thermal management, plasma diffusion, etc.). The control task requires simultaneously cooking three food items at distinct temperatures using two heat sources, posing challenges for policy learning under discretization constraints. We train Proximal Policy Optimization (PPO) agents across a Cartesian product of spatial resolution ratios $K$ and temporal resolution parameters $\log_{h_w}(h_r)$, yielding 25 experimental configurations as detailed in Figure 6. All experiments run on identical hardware (Intel Xeon Gold 6530, NVIDIA RTX 4090). Complete environment specifications, PPO hyperparameters, and convergence criteria are provided in Appendix E.

Figure 6 illustrates the comprehensive experimental results on the Teppanyaki thermal control task. Panels (a) and (b) present the environment setup: the agent regulates distributed heat sources to manage non-stationary diffusion dynamics for precise multi-object temperature tracking. Panel (c) displays $\varepsilon$-N scaling curves for all 23 qualified configurations, where color denotes temporal resolution $y$ and intensity indicates spatial resolution $K$. Panels (d), (e), and (f) show CPU, GPU, and total computational cost heatmaps across the $(K, y)$ configuration space. Configurations at $K = 8.0$

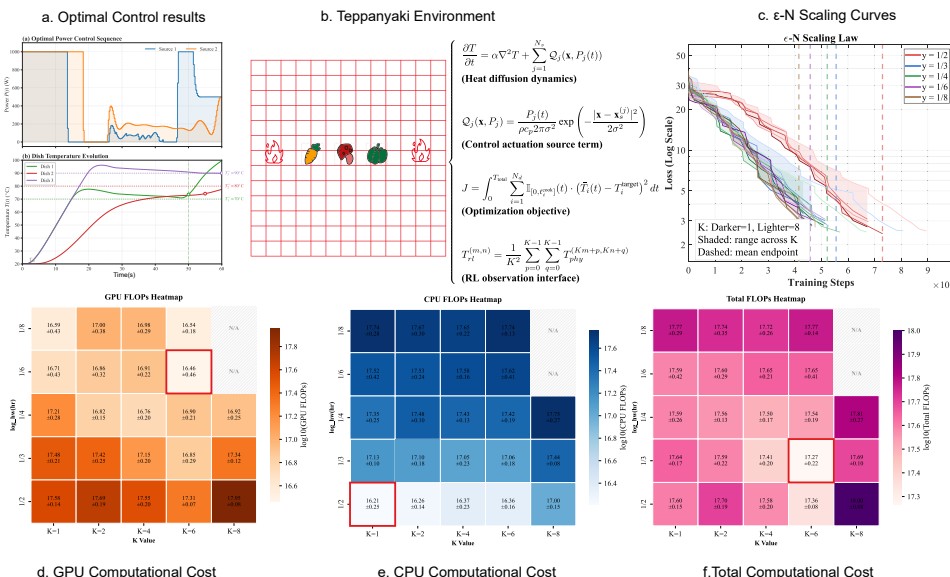

Figure 6: Experimental analysis of the Teppanyaki PDE control task. (a) The analytical optimal control solution serves as a performance oracle, illustrating the precise temperature tracking required. (b) Visualization of the simulation environment and the governing MHD/heat diffusion equations. (c-e) $\varepsilon$-N scaling curves (top) with CPU/GPU computational cost decomposition heatmaps (bottom), demonstrating the non-monotonic resolution-efficiency trade-off.

(marked "N/A") fail to converge, validating that excessive coarsening violates learnability conditions. The optimal configuration ($K^* = 6$, $\log_{h_w}(h_r) = 1/3$) achieves minimum total cost, confirming that Theorem 3 provides effective guidance for deep RL hyperparameter selection.

## 5 CONCLUSION AND LIMITATIONS

This work presents a unified theoretical framework for analyzing RL in surrogate-based physical environments governed by PDEs. By integrating spectral discretization theory with PAC sample complexity bounds from tabular RL, we establish how surrogate resolution, physical dynamics, and RL precision jointly influence the number of episodes required for accurate policy learning.

We propose the $\varepsilon-N$ framework to characterize the minimal RL–AI4S interaction cost under fixed accuracy and confidence requirements. Through analysis of surrogate-induced error propagations, and optimal resolution trade-off, we derive system-specific cost scaling laws and identify spectral regimes where surrogate refinement provides diminishing returns. Empirical validation across four representative AI4S-RL systems confirms that spectral growth rate, boundary observability, and dimensionality all play critical roles in determining learning efficiency.

Several assumptions in our analysis warrant further investigation. First, our analysis focuses on tabular RL for theoretical tractability and to isolate physical discretization errors, establishing a necessary condition for learnability that applies to any algorithm. While our experiments with DQN and PPO confirm that these resolution trade-offs govern the primary performance trends even in deep RL, rigorously extending our mathematical bounds to function approximation settings remains an open direction. Second, we treat surrogate models as fixed and externally trained; adaptive refinement or active error correction within the RL loop could further improve performance. Finally, our framework focuses on deterministic discretization errors. While real-world neural surrogates also introduce approximation and generalization errors, our analysis establishes a theoretical baseline , isolating the resolution trade-off as a fundamental constraint.

Overall, this work provides a principled foundation for evaluating and co-designing AI4S surrogates and RL policies in resource-constrained scientific applications.

ACKNOWLEDGMENTS

Kairui Feng was supported by the National Natural Science Foundation of China (Grant No. 62088101), Shanghai Municipal Science and Technology Major Project (Grant No. 2021SHZDZX0100), Explorer Program (Grant No. 24TS1401600), and Xiaomi Foundation.

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

## A    TABLE OF NOTATIONS

The following table provides a reference for the key symbols and notations used throughout this paper.

| Symbol | Description |
|---|---|
| **General Reinforcement Learning (RL) Parameters** | |
| $\mathcal{S}, \mathcal{A}$ | The finite sets representing the state space and action space. |
| $H$ | The horizon, representing the length of an episode. |
| $|\mathcal{S}|, |\mathcal{A}|$ | The size of the state and action spaces. |
| $\varepsilon$ | The desired error tolerance for value function estimation. |
| $\delta$ | The confidence parameter. |
| $V^\pi(s)$ | The value function of a policy $\pi$ at state $s$. |
| $V^*$ | The optimal value function. |
| $\pi^*, \hat{\pi}$ | The optimal policy and the learned policy, respectively. |
| $P, \hat{P}$ | The true and the empirically estimated state transition kernels. |
| $N$ | The sample complexity. The overall computational cost is treated as a function of $N$ in our $\varepsilon$-$N$ analysis. |
| **AI4S-RL Framework Parameters** | |
| $S_r, A_r, H_r$ | The scale of the state space, action space, and horizon in the RL domain. |
| $S_w, A_w, H_w$ | The scale of the state space, action space, and horizon in the AI4S domain. |
| $\Delta t_r, \Delta t_w$ | The discrete timesteps in the RL and AI4S/World domains, respectively. |
| $\Delta x_r, \Delta x_w$ | The spatial grid resolution in the RL and AI4S/PDE domains. |
| $K$ | The Resolution Ratio, defined as $K = \Delta x_r / \Delta x_w$. |
| $K^*$ | The optimal resolution ratio that minimizes computational cost. |
| $y_0, \Delta y_0$ | An initial state and the measurement uncertainty in practice. |
| $\Delta y$ | The grid cell size. |
| $\Delta_{total}$ | The total numerical error accumulated in one RL-AI4S interaction step. |
| $\rho$ | The forward projection error rate in the AI4S-RL system. |
| **PDE and Spectral Analysis Parameters** | |
| $\lambda_1$ | The dominant (largest real part) eigenvalue of the linearized PDE operator. |
| $\mathcal{L}(y_0)$ | The Fréchet-linearized operator of the PDE system around a state $y_0$. |
| $\phi_k$ | The $k$-th orthonormal basis function. |
| $\gamma_k(t)$ | The modal gain factor for the $k$-th mode at time $t$. |
| $\psi_k(t)$ | The propagated mode shape for the $k$-th mode at time $t$. |
| **Mathematical Spaces and Operators** | |
| $H^s(\Omega)$ | The Sobolev space of order $s$ over a domain $\Omega$. |
| $L^p(\Omega)$ | The Lebesgue space of $p$-integrable functions over a domain $\Omega$. |
| $\nabla, \Delta$ | The gradient and Laplacian operators, respectively. |
| $||\cdot||_V$ | The norm in a vector space $V$. |
| $\mathcal{D}f(y)[\eta]$ | The Fréchet derivative of an operator $f$ at point $y$ applied to a perturbation $\eta$. |

## B    SAMPLE COMPLEXITY IN REINFORCEMENT LEARNING

In this section, we provide an explanation of sample complexity in reinforcement learning. In our setting, the next-state information required by the reinforcement learning process is provided by a PDE-based physical space. We assume that this physical space satisfies the property of probabilistic determinability of the future state, meaning that, as long as repeated experiments are sufficiently performed, the next state can be identified with probability at least one. We present a detailed explanation of the probabilistic determinability of the future state.

## B.1 DIRECTIONAL PERTURBATION IN A NONLINEAR DYNAMICAL SYSTEM

To understand uncertainty propagation in complex dynamical systems, consider a nonlinear system governed by a PDE-based evolution operator $f(y_0, a)$, where $y_0$ is the initial state and $a$ denotes system parameters or external inputs. We focus on a single dimension (or component) of the system state, say $y^{(i)}$, and analyze how small perturbations in the initial condition affect the predicted future value $y_1^{(i)}$. Suppose that we do not observe $y_0$ exactly, but only a perturbed version $y_0 + \Delta y_0$, where $\|\Delta y_0\| \leq \Delta y$ reflects measurement uncertainty or modeling noise. A natural question arises:

*What is the possible range of future values $y_1^{(i)}$ under this uncertainty?*

A conventional linearized approximation assumes that the system's response to perturbations is symmetric and smooth around $y_0$. Under this assumption, we may apply a first-order Fréchet expansion

$$f(y_0 + \Delta y_0, a) \approx f(y_0, a) + \mathcal{D}f(y_0)[\Delta y_0],$$

and estimate the extreme values via projection onto the steepest ascent and descent directions. However, for highly nonlinear systems, this linear treatment is inadequate—particularly when the system exhibits directional sensitivity, i.e.,

$$f(y_0 + \eta, a) - f(y_0, a) \neq f(y_0 - \eta, a) - f(y_0, a).$$

To capture such asymmetry and nonlinear amplification, we adopt a variational formulation. Let the system's response to a perturbation path $\eta$ be described by a nonlinear variational functional:

$$\mathcal{I}[\eta] = \int_0^1 \mathcal{L}(y_0 + s\eta, \dot{\eta}, s)\, ds,$$

where $\mathcal{L}$ is a nonlinear Lagrangian that encodes the local effect of the perturbation path $\eta$ and its derivative $\dot{\eta}$ over time or space. The structural asymmetry of the system implies: $\mathcal{I}[\eta] \neq \mathcal{I}[-\eta]$, highlighting that perturbations in opposite directions produce asymmetric responses.

We define the extremal range of the future value $y_1^{(i)}$ as the solution to the following constrained nonlinear variational optimization:

$$
\begin{aligned}
y_1^{(i),\max} &= f^{(i)}(y_0) + \sup_{\|\eta\| \leq \Delta y} \left\{ \int_{\mathcal{M}} F^{(i)}[y_0, \eta(x), \nabla\eta(x)]\, d\mu(x) \right\}, \\
y_1^{(i),\min} &= f^{(i)}(y_0) + \inf_{\|\eta\| \leq \Delta y} \left\{ \int_{\mathcal{M}} F^{(i)}[y_0, \eta(x), \nabla\eta(x)]\, d\mu(x) \right\},
\end{aligned}
\tag{B.1}
$$

where $F^{(i)}$ denotes the nonlinear directional sensitivity of the system output with respect to the $i$-th component, and $\mu(x)$ is a measure over the spatial domain $\mathcal{M}$.

**Remark B.1.** *While this formulation is presented for a single component $y^{(i)}$, it naturally generalizes to the full system state vector. This variational perspective offers a principled and scalable method for quantifying directional uncertainty in nonlinear, asymmetric dynamical systems, with broad applicability to PDE-based forecasting, control, and decision-making.*

As illustrated in Figure B.1, in the ODE case, the three blue points represent the lower bound, true value, and upper bound of the observed state respectively, resulting from observation noise. Since substituting different observed states into the control equation yields different next-step predicted states, the three purple points correspond to the lower bound, true value, and upper bound of the next-step state.

We define a prediction to be correct if it falls into the same grid cell as the true next-step state. This correct cell is represented by the purple box. Predictions that fall into other grid cells are considered incorrect.

The PDE example in Figure B.1 illustrates how observation noise in the initial state propagates to the next time step. Here, we show the evolution over a single time step. The red dashed rectangle represents the range of the observed initial state induced by observation noise. The use of a rectangular region stems from modeling the perturbation $\delta y$ using the $L^\infty$ norm, which captures worst-case

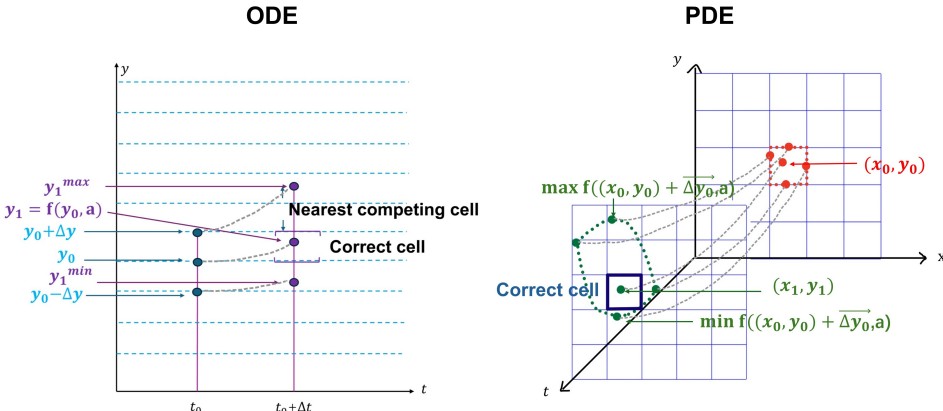

Figure B.1: A visual explanation of the probabilistic determinability of the future state.

uncertainty across all spatial dimensions. Under the assumption that the PDE solution operator is continuous with respect to the initial state in an appropriate Banach space, the open mapping theorem ensures that the reachable set of next-step states is also a topologically open and smooth manifold segment.

To further characterize the image of this noisy initial set under PDE evolution, we adopt a variational framework. Specifically, we consider a scalar output functional $\mathcal{F}[y]$ , and examine its extremal values under admissible perturbations $\delta y \in \mathcal{U}$, where $\mathcal{U} \subset T_{y_0}\mathcal{M}$ is a local uncertainty set defined on the tangent space of the solution manifold at $y_0$.

Let the PDE solution operator be denoted $y(t) = \mathcal{S}_t[y_0]$, where $y_0$ is the initial condition and $y(t)$ is the state at time $t$. For a scalar output functional $\mathcal{F}[y]$ (e.g., wave peak or energy), our goal is to evaluate how $\mathcal{F}[\mathcal{S}_{\Delta t}[y_0]]$ varies under perturbations $\delta y \in \mathcal{U} \subset T_{y_0}\mathcal{M}$, where $\mathcal{U}$ is a norm-bounded uncertainty set on the tangent space of the state manifold at $y_0$.

Formally, the extremal values of the output functional under admissible perturbations are given by the variational optimization:

$$\mathcal{F}_{\max} = \sup_{\delta y \in \mathcal{U}} \mathcal{F}[\mathcal{S}_{\Delta t}[y_0 + \delta y]],$$
$$\mathcal{F}_{\min} = \inf_{\delta y \in \mathcal{U}} \mathcal{F}[\mathcal{S}_{\Delta t}[y_0 + \delta y]].$$

This general variational formulation allows for full nonlinear and directional sensitivity to be incorporated. However, to derive computable approximations and gain insight into asymmetry, we next truncate the variation up to second order.

As an illustrative example, consider the one-dimensional wave equation:

$$\partial_t^2 y = c^2 \partial_x^2 y, \quad y(x,0) = y_0(x), \quad \partial_t y(x,0) = 0.$$

Let the output functional be the solution at a fixed point and time: $\mathcal{F}[y] = y(x^*, \Delta t)$. We expand the perturbed output $\mathcal{F}[y_0 + \delta y]$ to second order:

$$\mathcal{F}[y_0 + \delta y] = \mathcal{F}[y_0] + \int_0^L K(x)\delta y(x)\, dx + \frac{1}{2}\iint_{[0,L]^2} H(x, x')\delta y(x)\delta y(x')\, dx\, dx' + \mathcal{O}(\|\delta y\|^3),$$

where $K(x)$ is the first-order influence kernel, and $H(x, x')$ is the second-order interaction kernel.

The first-order term is linear and sign-symmetric: it leads to the same magnitude of change whether $\delta y$ is positive or negative. However, the second-order term breaks this symmetry. In particular, it captures nonlinear amplification and asymmetry: $\mathcal{F}[y_0 + \delta y] - \mathcal{F}[y_0] \neq -(\mathcal{F}[y_0 - \delta y] - \mathcal{F}[y_0])$, because the second-order term remains positive under sign reversal.

Under the constraint $\|\delta y\|_{L^\infty} \leq \varepsilon$, we then compute bounds:

$$
\begin{aligned}
\mathcal{F}_{\max} &= \mathcal{F}[y_0] + \sup_{\|\delta y\|_\infty \leq \varepsilon} \left\{ \int K(x)\delta y(x)\,dx + \frac{1}{2} \iint H(x,x')\delta y(x)\delta y(x')\,dx\,dx' \right\}, \\
\mathcal{F}_{\min} &= \mathcal{F}[y_0] + \inf_{\|\delta y\|_\infty \leq \varepsilon} \left\{ \int K(x)\delta y(x)\,dx + \frac{1}{2} \iint H(x,x')\delta y(x)\delta y(x')\,dx\,dx' \right\}.
\end{aligned}
\tag{B.2}
$$

This example demonstrates how second-order variational expansion serves as a concrete realization of the general nonlinear variational framework. It explicitly captures the directional asymmetry and curvature of the functional response to perturbations, which are essential features of PDE-governed systems with uncertain initial conditions.

## B.2 SAMPLING SCHEME FOR IDENTIFYING THE TRUE DISCRETE CELL

In this section, we analyze how a sampling strategy can be designed to identify the correct discrete cell that the system state should evolve into, and how the scale of such a cell should be chosen.

In real-world systems, perturbations $\Delta y$ typically originate from measurement noise or numerical rounding errors, and are therefore inevitable. Nevertheless, it is reasonable to assume that the true state lies within a neighborhood of size $\Delta y/2$ around the observed value. Based on this fact, we propose a local sampling strategy: perturbations are generated around the observed value within this neighborhood, and their evolution over one time step is computed via numerical PDE solvers. The resulting predicted outcomes are then statistically analyzed to infer the most probable cell in which the true state lies.

In the ideal noiseless case, such a sampling scheme causes predicted outcomes to concentrate within the correct cell. As the number of samples increases, the probability of correctly identifying the target cell converges to one.

This idea naturally supports the integration of physical knowledge into reinforcement learning. By repeatedly sampling perturbations around the initial state and propagating them through the physical model, we can estimate the state transition probability matrix with high confidence. This effectively transforms the reinforcement learning problem from a model-free setting to a dynamic programming problem with known transitions. In contrast, without such sampling, the system dynamics remain unknown, and exploration-based algorithms must be employed to estimate both the model and the optimal policy.

To ensure probabilistic determinability of the future state, we impose the following criterion: the number of predicted samples that fall into the correct cell must exceed that of any incorrect cell. Specifically, if the predicted uncertainty spans the correct cell and its two immediate neighbors, and if the correct cell is slightly larger in volume than its neighbors, then the correct cell can be statistically identified with high confidence. However, if one of the neighboring cells (e.g., the upper cell) is significantly larger than the others, then it becomes necessary to ensure that the number of predicted samples falling into the correct cell still exceeds those falling into the largest competing cell. This places stricter constraints on the acceptable error magnitude $\Delta y$, or alternatively, requires more samples to suppress statistical variance.

To improve sampling efficiency, we further propose a variationally-derived optimal importance sampling strategy. When the goal is to estimate the expected value of a functional output $\mathcal{F}(y_0 + \delta y)$, importance sampling theory shows that the optimal sampling distribution $p^*(\delta y)$ should satisfy $p^*(\delta y) \propto |\mathcal{F}(y_0 + \delta y)| \cdot \rho(\delta y)$, where $\rho(\delta y)$ is the base distribution of uncertainty. In the small-perturbation regime, we expand $\mathcal{F}(y_0 + \delta y)$ using a second-order Taylor approximation:

$$
\mathcal{F}(y_0 + \delta y) \approx \mathcal{F}(y_0) + \langle \nabla \mathcal{F}, \delta y \rangle + \frac{1}{2}\delta y^\top H \delta y,
$$

leading to the approximate optimal sampling density $p^*(\delta y) \propto \left|\langle \nabla\mathcal{F}, \delta y \rangle + \frac{1}{2}\delta y^\top H \delta y\right|$. Accordingly, we define a sampling weight function: $w(\delta y) := \left|\langle \nabla\mathcal{F}, \delta y \rangle + \frac{1}{2}\delta y^\top H \delta y\right|$, which can be used to guide sampling toward directions with the most significant influence on the predicted outcome. This minimizes estimation variance and achieves optimal sampling efficiency. The resulting scheme

can be interpreted as a physics-informed importance sampling strategy, grounded in variational sensitivity analysis and importance sampling theory.

We now quantify how many samples are needed to reliably identify the correct discrete cell when using the optimal importance sampling strategy described previously. Specifically, we define the *dominant cell* as the cell that receives the largest number of predicted samples under a given sampling scheme. Let $C^*$ denote the true (correct) cell and $C_j$ denote any competing cell. Let $\pi^* := \mathbb{P}[\mathcal{F}(y_0 + \delta y) \in C^*]$ and $\pi_j := \mathbb{P}[\mathcal{F}(y_0 + \delta y) \in C_j]$, where $\delta y \sim p^*(\delta y) \propto |\mathcal{F}(y_0 + \delta y)| \cdot \rho(\delta y)$ is the optimal importance sampling distribution.

**Theorem B.1** (Sample Complexity for Correct Cell Dominance)**.** *Let $K$ be the total number of cells, and let $\delta \in (0, 1)$ be the desired confidence level. Assume that $\Delta := \pi^* - \pi_{\max} > 0$, where $\pi_{\max} = \max_{j \neq C^*} \pi_j$. Then, in order for the correct cell $C^*$ to receive the largest number of samples with probability at least $1 - \delta$, it suffices to sample:*

$$N \geq \frac{8}{\Delta^2} \cdot \log\left(\frac{2(K-1)}{\delta}\right)$$

*samples from the optimal importance sampling distribution $p^*(\delta y)$.*

*Proof.* Let $n_{C^*} \sim \text{Binomial}(N, \pi^*)$ and $n_j \sim \text{Binomial}(N, \pi_j)$ for each $j \neq C^*$. To ensure that $n_{C^*} > n_j$, we use a union bound and Chernoff-type large deviation inequality. The error probability that any incorrect cell receives more samples than $C^*$ is bounded as:

$$\mathbb{P}\left(\exists j \neq C^* : n_j \geq n_{C^*}\right) \leq 2(K-1) \cdot \exp\left(-\frac{N\Delta^2}{8}\right).$$

Setting the right-hand side less than or equal to $\delta$ yields the desired bound on $N$. $\square$

This result shows that the sample complexity scales inversely with the square of the gap $\Delta = \pi^* - \pi_{\max}$ between the correct cell and the nearest competitor, and logarithmically with the number of cells and the inverse failure probability. This quantifies how distinguishable the correct cell is under the importance sampling distribution. The smaller the margin $\Delta$, the more samples are needed to overcome statistical uncertainty.

However, to preserve generality and narrative consistency, we adopt a uniform sampling strategy in the main analysis to estimate sample complexity, which differs from the optimal importance sampling scheme only by a constant-factor overhead.

### B.3    PROBLEM FORMULATION

We consider two distinct scenarios concerning the availability of transition dynamics. These two cases correspond to two different learning regimes, each with its own complexity characteristics.

In the following analysis, we quantify the sample complexity under both settings. Based on Lemma B.1 and Theorem B.2, we derive two different orders of sample complexity. The bound in Lemma B.1 captures the general case without prior knowledge of the transition dynamics. In contrast, Theorem B.2 establishes the bound under our setting, where the transition matrix is known through repeated sampling from the PDE-based physical space.

Therefore, if no sampling is performed in the physical space, the problem remains a standard MDP, and exploration is required to learn the model. However, if sufficient sampling is carried out in the PDE domain, the problem effectively becomes a dynamic programming task, and the sample complexity is driven by the cost of interaction with the physical model, resulting in a tighter bound. In the main body of the paper, we adopt the sample complexity bound given by Theorem B.2. In the following, we present several standard assumptions commonly used in the conventional MDP framework.

**Markov Decision Problems** We consider episodic reinforcement learning in finite-horizon MDPs defined by $< S, A, P, R, H >$, where $S$ and $A$ are the finite sets of states and actions, $P$ is the state transition distribution, the function $R$ is a real-valued function which is deterministic and belongs to the interval $[0, 1]$, and the horizon $H$ is the length of the episode. We denote by $P(\cdot|x, a)$ and $R(x, a)$

the probability distribution over the next state and the immediate reward of taking action $a$ at state $x$, respectively.

The agent interacts with the environment in a sequence of episodes. Specifically, the agent follows a policy $\pi : S \times [H] \to A$, which maps each state and time step to an action. The value function $V_h^\pi : S \to \mathbb{R}$ represents the expected cumulative reward from step $h$ to $H$ under policy $\pi$, starting from state $x$ at time $h$. Under the previously defined MDP setting, there always exists an optimal policy $\pi^*$ attaining the optimal value function $V_h^*(x) \triangleq \sup_\pi V_h^\pi(x)$ for all $x \in S$ and $h \in [H]$.

Each policy $\pi$ induces a transition kernel $P_h^\pi(y|x) \triangleq P(y|x, \pi(x,h))$ and reward function $r_h^\pi(x) \triangleq R(x, \pi(x,h))$. For any function $V : S \to \mathbb{R}$, we define the transition operators: $(P_h^\pi V)(x) \triangleq \sum_{y \in S} P_h^\pi(y|x) V(y)$. The Bellman operator for policy $\pi$ is defined as $(\mathcal{T}_h^\pi V)(x) \triangleq r_h^\pi(x) + (P_h^\pi V)(x)$.

We assume that the reward function $r$ is known to the agent but the transition kernel $p$ is unknown (Under the no-sampling setting). The question we study is how many episodes does a learning agent follow a policy $\pi$ that is not $\epsilon$-optimal, i.e., $V_M^* - \epsilon > V_M^\pi$, with probability at least $1 - \delta$ for any chosen accuracy $\epsilon$ and failure probability $\delta$.

### B.4 FULL ANALYSIS OF SAMPLE COMPLEXITY BOUND

In this section, we systematically analyze the PAC sample complexity of reinforcement learning under two different assumptions regarding the transition probability matrix. We begin by examining the classical setting where the transition dynamics are entirely unknown. Lemma B.1 provides a baseline PAC bound in this setting, and reveal how the total sample complexity arises from key error components, including estimation error, exploration bonuses, and confidence adjustments.

Building on this foundation, we then analyze how our proposed setting—where additional structure or prior knowledge is introduced—modifies the learning process. In particular, Lemmas B.2 and B.3 show how specific components in the original analysis are tightened or avoided, leading to an improved sample complexity bound.

**Theorem B.2** (PAC Bound under our setting). *Assume the transition dynamics of the MDP are known up to statistical confidence level $\delta$, reducing the reinforcement learning problem to dynamic programming. Let the correct classification cell have predicted frequency $p = \Delta y^{(j)}$, and let the most probable competing cell have frequency $q = p_{\max}^{(j)}$. Then the number of forward predictions required to identify the correct classification cell with probability at least $1 - \delta$ satisfies*

$$n = \mathcal{O}\left( HSA \cdot \frac{\log(1/\delta)}{\min_j \left( \Delta y^{(j)} - p_{\max}^{(j)} \right)^2} \right).$$

*Furthermore, we derive a refined bound on the sample complexity $n = \mathcal{O}\left( \frac{SAH^3}{\varepsilon^2} \cdot \log(1/\delta) \right)$.*

We also present in Lemma B.1 the PAC bound of the classical UCB-VI algorithm under the setting where the state transition probability matrix is entirely unknown. The detailed algorithmic procedure of UCB-VI can be found in (Azar et al., 2017).

**Lemma B.1.** *For any $0 < \epsilon, \delta \leq 1$, the following holds. With probability at least $1 - \delta$, the algorithm $UCB-VI$ produces a sequence of policies $\pi^k$, that yield at most $\mathcal{O}\left( \frac{SAH^3}{\varepsilon^2} \log \frac{1}{\delta} \right)$ episodes where $V_{1:H}^*(s_0) - V_{1:H}^{\pi^k}(s_0) > \epsilon$. This results in a total runtime for sampling of $\mathcal{O}\left( \frac{SAH^4}{\varepsilon^2} \log \frac{1}{\delta} \right)$.*

*Proof.* Lemma B.1 characterizes the PAC bound of the UCB-VI algorithm, a result that has been rigorously established and thoroughly analyzed in prior work (see Azar et al. (2017)). To facilitate our later analysis, we provide a concise overview of the proof from Azar et al. (2017), with particular emphasis on the error decomposition framework and the core techniques used to bound individual error components.

Let $\mathcal{V} = \{V_{k,h} \geq V_h^*, \forall k, h\}$ denote the event where all computed value estimates are optimistic upper bounds of the optimal value function. Azar et al. (2017) show that $\mathcal{V}$ holds with probability

at least $1 - \delta$ under standard induction arguments with appropriate bonus functions. Define the approximation errors $w_{k,h} = W_{k,h}(x_{k,h})$ and $\tilde{w}_{k,h} = \tilde{W}_{k,h}(x_{k,h})$, where $W_{k,h} = V_h^* - V_h^{\pi_k}$ and $\tilde{W}_{k,h} = V_{k,h} - V_h^{\pi_k}$. Then, under the event $\mathcal{V}$, the following error decomposition holds:

$$w_{k,h} \leq \tilde{w}_{k,h} = \mathcal{O}\left(\sum_{i=h}^{H-1}\left(M_{k,i} + 2\sqrt{L}\bar{M}_{k,i} + \text{bonus}_{k,i} + E_{k,i}\right)\right),$$

where $M_{k,h} = P^{\pi_k}W_{k,h+1}(x_{k,h}) - W_{k,h+1}(x_{k,h+1})$ is a martingale difference term. $\bar{M}_{k,i}$ is the quantity resulting from applying Bernstein's inequality to control the term $\left[(\hat{P}_h^\pi - P_h^\pi)(V_{h+1} - V_{h+1}^*)\right](x_h)$ and is also a martingale difference. $\text{bonus}_{k,i}$ represents the bonus function for UCB-VI at step $i$ under episode $k$. $E_{k,h} \stackrel{\text{def}}{=} (\hat{P}_k^{\pi_k} - P^{\pi_k})V_{h+1}^*(x_{k,h})$ is the estimation error of the optimal value function at the next state. In addition, $L$ here is a logarithmic factor depending on $H$, $S$, $A$, and the confidence level $\delta$, which does not affect the polynomial complexity of the bound.

Based on the above statement, we observe that the UCB-VI algorithm decomposes the total error into four components: $M_{k,i}$, $\bar{M}_{k,i}$, $\text{bonus}_{k,i}$, and $E_{k,i}$. In Table B.2, we summarize the order of magnitude and the bounding techniques for each of these four error terms. The detailed proof process can be found in (Azar et al., 2017). Therefore, the cumulative value error over all episodes and timesteps satisfies $\sum_{k,h} w_{k,h} = \mathcal{O}\left(HL\sqrt{SAT}\right)$, where $T = KH$ is the total number of steps collected across $K$ episodes, each of length $H$. This implies:

$$\sum_{k,h} w_{k,h} = \mathcal{O}\left(HL\sqrt{SAKH}\right) = \mathcal{O}\left(H^{3/2}L\sqrt{SAK}\right).$$

Suppose in the worst-case scenario, all of the first $K$ episodes are not $\varepsilon - optimal$ before an $\varepsilon - optimal$ policy is learned, then we require:

$$K_{\text{bad}} \cdot \varepsilon \leq \sum_{k,h} w_{k,h} = \mathcal{O}\left(H^{3/2}L\sqrt{SAK_{bad}}\right) \Rightarrow K_{\text{bad}} \leq \mathcal{O}\left(\frac{H^3L^2SA}{\varepsilon^2}\right) = \mathcal{O}\left(\frac{SAH^3}{\varepsilon^2}\log\frac{1}{\delta}\right).$$

To derive the total sample complexity, we note that sampling one episode and updating the respective variables has $\mathcal{O}(H)$ runtime. Thus, the total runtime across all $K_{\text{bad}}$ episodes is $\mathcal{O}\left(H \cdot K_{\text{bad}}\right) = \mathcal{O}\left(\frac{SAH^4}{\varepsilon^2}\log\frac{1}{\delta}\right)$. This establishes the total sample complexity required to ensure that the learned policy is $\varepsilon$-optimal with high probability.

Table B.2: Error decomposition in UCB-VI and bounding techniques

| Error Term | Magnitude | Bounding Technique |
|---|---|---|
| $\sum_{k,h} M_{k,h}$ | $\mathcal{O}(H\sqrt{TL})$ | Azuma's inequality (martingale bound) |
| $\sum_{k,h} \bar{M}_{k,h}$ | $\mathcal{O}(\sqrt{TL})$ | Azuma's inequality (martingale bound) |
| $\sum_{k,h} E_{k,h}$ | $\mathcal{O}(H\sqrt{LSAT})$ | Chernoff-Hoeffding inequality + pigeonhole principle |
| $\sum_{k,h} \text{bonus}_{k,h}$ | $\mathcal{O}(HL\sqrt{SAT})$ | Optimistic bonus design + pigeonhole principle |

$\square$

**Remark B.2.** *Based on Table B.2, we have essentially clarified the analytical procedure through which the UCB-VI algorithm derives its sample complexity bound. Based on this, we highlight the key difference between our setting and UCB-VI, and the core improvement introduced in our approach.*

*UCB-VI characterizes sample complexity by bounding the discrepancy between the empirical transition probability matrix obtained from sampling and the true transition dynamics.*

*In contrast, our setting introduces a notion of the probabilistic determinability of future state, which allows us to exploit multiple samples from the PDE-based physical space to ensure that the transition probability matrix in the RL formulation is fully known.*

*As a result, the increase in sample size in our framework pertains to the sampling process in the PDE-based physical space. The complexity of this sampling process can then be transferred to the RL domain, ultimately yielding a refined bound on sample complexity.*

In Lemma B.2, we provide the sampling requirement needed to resolve the correct cell with probability at least $1 - \delta$.

**Lemma B.2** (Sample Complexity for $\delta$-Confidence Classification)**.** *Let the cell of correct classification have predicted frequency $p = \Delta y^{(j)}$, and let the most probable incorrect (competing) cell have frequency $q = p_{\max}^{(j)}$. To resolve the correct cell with confidence at least $1 - \delta$, the number of forward predictions required satisfies*

$$n = \mathcal{O}\left(\frac{\log(1/\delta)}{\min_j \left(\Delta y^{(j)} - p_{\max}^{(j)}\right)^2}\right).$$

*Proof.* Without loss of generality, we consider a randomly selected dimension $j$ as an example. Let $p = \Delta y^{(j)}$ be the probability mass of the correct cell in dimension $j$, and let $q = p_{\max}^{(j)}$ be the maximum probability mass of any competing cell.

Suppose we sample $n$ i.i.d. trajectories, and let $\hat{p}$ denotes the empirical frequencies of the correct cell. We wish to ensure that with high probability (at least $1 - \delta$), $|\hat{p} - p| < \frac{|p-q|}{2}$, so that the correct classification cell is selected.

Let $\epsilon = \frac{|p-q|}{2}$. Applying Hoeffding's inequality to $\hat{p}$ gives:

$$\Pr\left(|\hat{p} - p| \geq \epsilon\right) \leq 2\exp(-2n\epsilon^2) \Rightarrow \Pr\left(|\hat{p} - p| \geq \frac{|p-q|}{2}\right) \leq 2\exp(-2n(\frac{|p-q|}{2})^2).$$

Then enforce $2\exp\left(-2n\left(\frac{p-q}{2}\right)^2\right) \leq \delta \Rightarrow n \geq \frac{2\log(2/\delta)}{(p-q)^2}$.

This gives the number of samples $n$ needed to distinguish between the correct and competing cells with probability at least $1 - \delta$. To guarantee correct classification across all dimensions $j$, we must take the worst-case (smallest gap) over all $j$, yielding: $n = \mathcal{O}\left(\frac{\log(1/\delta)}{\min_j \left(\Delta y^{(j)} - p_{\max}^{(j)}\right)^2}\right)$ as stated. $\square$

Furthermore, Lemma B.3 provides the order of the total sample complexity under our setting.

**Lemma B.3** (Refined Sample Complexity under Known Transitions)**.** *Assume the transition dynamics of the MDP are known up to statistical confidence level $\delta$, reducing the reinforcement learning problem to dynamic programming. Let the correct classification cell have predicted frequency $p = \Delta y^{(j)}$, and let the most probable competing cell have frequency $q = p_{\max}^{(j)}$. Then the number of forward predictions required to identify the correct classification cell with probability at least $1 - \delta$ satisfies*

$$n = \mathcal{O}\left(H_r S_r A_r \cdot \frac{\log(1/\delta)}{\min_j \left(\Delta y^{(j)} - p_{\max}^{(j)}\right)^2}\right).$$

*Furthermore, by applying the $\rho - K$ relationship discussed in the main text, which states that $\Delta p^{(j)} \sim \rho \Delta y^{(j)} \sim \frac{1}{H_r} \Delta y^{(j)}$, we derive a refined bound on the sample complexity:*

$$n = \mathcal{O}\left(H_r^3 S_r A_r \cdot \log(1/\delta)\right).$$

*Proof.* Without loss of generality, consider a randomly selected dimension $j$. Let $p = \Delta y^{(j)}$ be the predicted frequency of the correct classification cell, and let $q = p_{\max}^{(j)}$ denote the predicted frequency of the most likely competing cell in dimension $j$.

Since the transition matrix is assumed to be known up to confidence $\delta$, the reinforcement learning task reduces to a dynamic programming problem. Assume we sample $n$ independent forward trajectories and let $\hat{p}$ be the empirical frequency of the correct cell. To confidently resolve the correct classification, we want the empirical estimate to satisfy $|\hat{p} - p| < \frac{|p-q|}{2}$ with probability at least $1 - \delta$. Let $\epsilon = \frac{|p-q|}{2}$. According to Lemma B.2, we have $n \geq \frac{2\log(2/\delta)}{(p-q)^2}$.

At each step in the reinforcement learning process, every state-action pair requires the number of samples derived above to guarantee correct classification with high confidence. Applying a union bound over all such points yields a total sample complexity of:

$$n = \mathcal{O}\left( H_r S_r A_r \cdot \frac{\log(1/\delta)}{\min_j (p-q)^2} \right).$$

Furthermore, we have: $p - q \sim \mathcal{O}\left(\frac{1}{H_r}\right)$, thus $\frac{1}{(p-q)^2} = \mathcal{O}(H_r^2)$. Substituting this into the earlier bound yields: $n = \mathcal{O}\left( H_r^3 S_r A_r \cdot \log(1/\delta) \right)$ as claimed. $\qquad\square$

**Remark B.3.** *Based on Lemma B.1 and Theorem B.2, we derive two different orders of sample complexity. However, we also leave open the possibility for further refinement. In particular, building upon the analytical framework presented above, future research may explore combining our treatment of transition uncertainty with recent developments on PAC bounds in robust MDPs, potentially yielding tighter complexity guarantees.*

## C    DETAILED ANALYSIS OF THE FOUR REPRESENTATIVE AI4S-RL ENVIRONMENTS

In this section, building on the theoretical framework proposed in the main text, we provide a detailed analysis of how surrogate discretization, spectral response, and reinforcement learning precision jointly affect sample complexity across four representative AI4S-RL systems: (1) Tokamak plasma control, (2) Turbulent airfoil regulation, (3) Teppanyaki heat sequencing, and (4) Cart-pole stabilization. The first three environments are PDE-based systems, while the fourth represents a typical ODE-based system. Notably, the first two environments are characterized as systems with strong boundary observability. Therefore, by conducting a detailed discussion of these four environments, we analyze how different environmental characteristics and underlying equation types influence the outcomes and properties of optimal resolution matching and computational cost, thereby enhancing the generality of our conclusions.

### C.1    THEORETICAL PRELIMINARIES AND PROOFS

Before proceeding to the detailed analysis of each environment, this subsection provides several general settings and supplementary explanations for key lemmas and conclusions referenced in the main text. In addition, Appendix D provides a collection of classical results from functional analysis and PDE theory that underpin the error analysis.

#### C.1.1    PROOF OF THEOREM 1

**Theorem C.3** (Sample Complexity for $\delta$-Confidence Classification). *Consider repeated forward predictions from perturbed initial states, where the predicted frequency is the empirical probability of the next state falling into a given grid cell. Let $p$ denote the predicted frequency with which the true next state falls into the correct grid cell, and let $q = p_{\max}^{(j)}$ denote the maximum predicted frequency among all competing cells. To resolve the correct cell with confidence at least $1 - \delta$, the number of*

*forward predictions required is bounded by*

$$n = \mathcal{O}\left(\frac{\log(1/\delta)}{\min_j \left(\Delta y^{(j)} - p_{\max}^{(j)}\right)^2}\right) \tag{C.1}$$

*Proof.* We proceed under the fundamental assumption of **state distinguishability**, which ensures that the propagation of initial uncertainty is sufficiently bounded. This allows for the correct classification cell to be identified, as repeated sampling from the input perturbation ball will generate predicted states that form a distinct cluster primarily within that cell.

Consider $n$ independent forward predictions, each obtained by sampling a perturbation $\Delta y_0$ uniformly from the ball $\{\|\Delta y_0\| \leq \Delta y\}$ and computing $\tilde{y}_1 = f(y_0 + \Delta y_0, a)$. Let $X_i$ be the indicator random variable for the $i$-th prediction falling into the correct cell: $X_i = \begin{cases} 1 & \text{if } \tilde{y}_1^{(i)} \text{ falls in the correct cell} \\ 0 & \text{otherwise} \end{cases}$.

The empirical frequency of the correct cell is: $\hat{p} = \frac{1}{n}\sum_{i=1}^n X_i$. By construction, $\mathbb{E}[X_i] = p$ and $X_i \in [0, 1]$.

Since $X_1, X_2, \ldots, X_n$ are independent and bounded random variables with $X_i \in [0, 1]$, Hoeffding's inequality states: $\Pr(|\hat{p} - p| \geq \epsilon) \leq 2\exp(-2n\epsilon^2)$ for any $\epsilon > 0$. To distinguish the correct cell from competing cells, we need $\hat{p}$ to be closer to $p$ than to $q$. This is achieved if: $|\hat{p} - p| < \frac{|p-q|}{2}$. Setting $\epsilon = \frac{|p-q|}{2} = \frac{p-q}{2}$ (since $p > q$ by assumption), we get:

$$\Pr\left(|\hat{p} - p| \geq \frac{p - q}{2}\right) \leq 2\exp\left(-2n\left(\frac{p - q}{2}\right)^2\right)$$

For the classification to succeed with confidence at least $1 - \delta$, we require:

$$\Pr\left(|\hat{p} - p| \geq \frac{p - q}{2}\right) \leq 2\exp\left(-2n\left(\frac{p - q}{2}\right)^2\right) \leq \delta$$

Taking logarithms and solving for $n$: $n \geq \frac{2\log(2/\delta)}{(p-q)^2}$. When multiple dimensions are involved, we need to ensure correct classification across all dimensions $j$. Taking the worst-case scenario (smallest gap):

$$n \geq \max_j \frac{2\log(2/\delta)}{(\Delta y^{(j)} - p_{\max}^{(j)})^2} = \frac{2\log(2/\delta)}{\min_j(\Delta y^{(j)} - p_{\max}^{(j)})^2}$$

In terms of asymptotic complexity (Big-O notation), the constant factors can be absorbed, yielding the final result: $n = \mathcal{O}\left(\frac{\log(1/\delta)}{\min_j(\Delta y^{(j)} - p_{\max}^{(j)})^2}\right)$. This completes the proof of the sample complexity bound. $\square$

### C.1.2 MODAL GROWTH RATE AND STATE SEPARABILITY

In AI4S-RL, agents interact with high-dimensional physical systems, where observations are often contaminated with sensor noise or state perturbations. A key concern is whether such uncertainty grow through the system's nonlinear dynamics and cause classification ambiguity in the predicted future state.

To ensure robustness of the learning process, we must guarantee that small observation errors do not cause the predicted future state $f(y_0 + \Delta y_0)$ to cross classification boundaries. This leads to the formulation of Lemma S.3, which characterizes sufficient conditions for achieving classification-cell separability.

**Lemma S.3** (Fourier-Mode Separability via Fréchet Spectral Amplification). *Let $f_t(y_0)$ be the solution operator of a nonlinear PDE with Fréchet derivative $\mathcal{D}f_t(y_0)$ acting on a Hilbert space $\mathcal{H}$. Suppose the observation perturbation $\eta \in \mathcal{H}$, with $\|\eta\| \leq \Delta y$, admits a Fourier expansion $\eta = \sum_k \hat{\eta}_k \phi_k$, where $\{\phi_k\}$ is an orthonormal basis for $\mathcal{H}$.*

*Assume the Fréchet derivative propagates each mode as $\mathcal{D}f_t(y_0)[\phi_k] = \gamma_k(t)\psi_k(t)$ for some gain factor $\gamma_k(t) > 0$. Then classification-cell separability is guaranteed if: $\sup_k \gamma_k(t) < 1$, which implies $\|\mathcal{D}f_t(y_0)[\eta]\| < \Delta y$. Consequently, the maximal admissible time step $\Delta t$ satisfies:*

$$\Delta t < C_{system} := \inf_k \gamma_k^{-1}(1).$$

*Proof.* Let $\eta = \sum_k \hat{\eta}_k \phi_k \in \mathcal{H}$ be the perturbation with norm $\|\eta\| \leq \Delta y$, where $\{\phi_k\}$ is an orthonormal basis. Since $\mathcal{D}f_t(y_0)$ is linear, then we have:

$$\mathcal{D}f_t(y_0)[\eta] = \sum_k \hat{\eta}_k \cdot \mathcal{D}f_t(y_0)[\phi_k] = \sum_k \hat{\eta}_k \cdot \gamma_k(t) \cdot \psi_k(t).$$

Now take the squared norm:

$$\|\mathcal{D}f_t(y_0)[\eta]\|^2 = \left\|\sum_k \hat{\eta}_k \cdot \gamma_k(t)\right\|^2 \leq \left(\sup_k \gamma_k(t)^2\right) \cdot \sum_k |\hat{\eta}_k|^2 = \sup_k \gamma_k(t)^2 \cdot \|\eta\|^2.$$

Therefore,

$$\|\mathcal{D}f_t(y_0)[\eta]\| \leq \sup_k \gamma_k(t) \cdot \|\eta\| \leq \sup_k \gamma_k(t) \cdot \Delta y.$$

To ensure $\|\mathcal{D}f_t(y_0)[\eta]\| < \Delta y$, it suffices that $\sup_k \gamma_k(t) < 1$. $\qquad\square$

The spectral separability condition established in Lemma S.3 applies broadly to a wide class of PDE-governed systems. In general, when the system dynamics are governed by a nonlinear evolution equation, local behavior around state $y_0$ can be analyzed through Fréchet linearization, yielding a time-dependent linear operator $\mathcal{D}f_t(y_0)$. By decomposing initial observation perturbations $\eta \in \mathcal{H}$ into orthonormal modal components and tracking the amplification of each mode under the action of $\mathcal{D}f_t(y_0)$, we obtain a spectral characterization of uncertainty propagation over time.

### C.1.3 Relating Modal Gain $\gamma_k(t)$ to the Leading Eigenvalue $\lambda_1$

In the framework discussed in the previous section, each mode experiences a gain factor $\gamma_k(t)$, which reflects the sensitivity of the system to perturbations in that direction. When the maximal gain $\sup_k \gamma_k(t)$ remains below unity, the propagated perturbation stays within the classification boundary, ensuring separability and decision consistency. This condition naturally induces a system-dependent constraint on temporal resolution: the RL decision step size must be sufficiently high to resolve the most unstable direction in the spectral space. In practice, this analysis enables principled derivation of time-step bounds for a variety of AI4S-RL environments, thereby enabling robust policy learning grounded in physical dynamics.

**Remark C.4.** *In nonlinear dynamical systems governed by PDEs, the rigorous characterization of forward perturbation growth should ideally be expressed in terms of the Fréchet-mode amplification factors $\gamma_k(t)$, as introduced in Lemma S.3. These quantities measure the directional sensitivity of the solution operator $f_t$ along each Fourier mode $\phi_k$, through the relation $\mathcal{D}f_t(y_0)[\phi_k] = \gamma_k(t)\psi_k(t)$, where $\psi_k(t)$ denotes the propagated mode shape. However, to simplify the analysis and maintain consistency across sections, we adopt the leading eigenvalue $\lambda_1$ of the linearized system operator $\mathcal{L}(y_0)$ as a surrogate for modal growth, particularly in the computation of time resolution bounds and error propagation rates. The relationship between $\lambda_k$ and $\gamma_k(t)$ can be made explicit under the assumption of locally linearized evolution:*

$$\delta y_k(t) = \hat{\eta}_k e^{\lambda_k t} \phi_k \quad \Rightarrow \quad \gamma_k(t) \propto e^{\lambda_k t}.$$

*For sufficiently small $t$, Taylor expansion yields: $\gamma_k(t) \approx 1 + \lambda_k t + \mathcal{O}(t^2)$, so that $\sup_k \gamma_k(t) \propto \lambda_1$. Therefore, replacing $\sup_k \gamma_k(t)$ with $\lambda_1$ in our main analysis introduces no change in asymptotic error scaling.*

### C.1.4 PROOF OF THEOREM 2

**Theorem C.4** ($\rho$-$K$ Analysis for $d$-Dimensional Systems). *For a $d$-dimensional PDE system, the forward projection error rate is:*

$$\rho = 1 - \frac{1}{\lambda_1/H_r + 1 + C_1/K^d} \tag{C.2}$$

*where the numerical error scales as $\Delta_{total} \sim C_1 K^{-d} \Delta y$ from spatial discretization. In the high-resolution limit:*

$$\rho = \mathcal{O}\left(\frac{1}{H_r} + \frac{1}{K^d}\right) \tag{C.3}$$

*Proof.* The forward projection error rate, $\rho$, is defined as the fractional deviation from a perfect prediction. Conceptually, this is one minus the ratio of the ideal state-cell size to the total spread of the predicted state under all sources of error. Let the ideal cell size be represented by the baseline observation uncertainty, $\Delta y$.

$$\rho = 1 - \frac{\text{Ideal Cell Size}}{\text{Total Error Spread}} = 1 - \frac{\Delta y}{\text{Total Error Spread}}$$

The total error spread is the linear superposition of three primary components identified in the text:

1. **Intrinsic Error Growth ($\Delta_{\text{intrinsic}}$):** The amplification of the initial uncertainty $\Delta y$ by the system's dynamics over a single RL time step $\Delta t_r = 1/H_r$. This is governed by the leading eigenvalue $\lambda_1$, yielding $\Delta_{\text{intrinsic}} = \frac{\lambda_1}{H_r} \Delta y$.

2. **Baseline Observation Uncertainty ($\Delta_{\text{obs}}$):** The inherent noise floor of the system, which is $\Delta y$.

3. **Numerical Surrogate Error ($\Delta_{\text{num}}$):** The total error from discretization, $\Delta_{\text{total}}$, which the theorem states scales as $\Delta_{\text{total}} \sim C_1 K^{-d} \Delta y$. This scaling arises from the resolution matching conditions, where the dominant contribution to error that depends on the resolution ratio $K$ is the spatial discretization of the underlying PDE in a $d$-dimensional space.

Summing these components and canceling the $\Delta y$ term yields the first result of the theorem:

$$\rho = 1 - \frac{1}{\frac{\lambda_1}{H_r} + 1 + C_1 K^{-d}}$$

The high-resolution limit is defined as the case where $H_r \to \infty$ and $K \to \infty$. To analyze the behavior of $\rho$ in this limit, we first rearrange the expression:

$$1 - \rho = \frac{1}{1 + \frac{\lambda_1}{H_r} + C_1 K^{-d}}$$

Let $x = \frac{\lambda_1}{H_r} + C_1 K^{-d}$. In the high-resolution limit, as $H_r \to \infty$ and $K \to \infty$, it is clear that $x \to 0$. We can therefore use the first-order Taylor series expansion for the function $f(x) = \frac{1}{1+x}$ around $x = 0$. Substituting our expression for $x$: $1 - \rho \approx 1 - \left(\frac{\lambda_1}{H_r} + C_1 K^{-d}\right)$. Then we obtain the approximate expression for $\rho$: $\rho \approx \frac{\lambda_1}{H_r} + C_1 K^{-d}$. Since $\lambda_1$ and $C_1$ are constants for a given system, the asymptotic behavior of $\rho$ is captured by Big-O notation as:

$$\rho = \mathcal{O}\left(\frac{1}{H_r} + \frac{1}{K^d}\right)$$

This completes the proof of the second part of the theorem. $\qquad\square$

### C.1.5 PROOF OF THEOREM 3

**Theorem C.5** (Optimal Resolution with System-Dependent Scaling). *For a physical system with state space scaling $S_r \sim K^\alpha$ and action space scaling $A_r \sim K^\beta$, under the computational balance condition $H_r^3 S_r A_r \sim H_w S_w A_w$, minimizing the computational cost*

$$Cost(K) = H_r^3 K^{\alpha+\beta} \cdot \left( \frac{\log(1/\delta)}{\varepsilon^2} \right) \cdot \left( \frac{1}{1 - \frac{1}{H_r} - \frac{1}{K^d}} \right)^2 \tag{C.4}$$

*yields optimal resolution:*

$$K^* = \left( \frac{\alpha + \beta + 2d}{(\alpha + \beta)(1 - H_r^{-1})} \right)^{1/d} \approx \left( \frac{\alpha + \beta + 2d}{\alpha + \beta} \right)^{1/d} \cdot \exp\left( \frac{1}{d\lambda_1} \right) \tag{C.5}$$

*when $H_r \gtrsim \lambda_1 \gg 1$.*

*Proof.* The total computational cost is given by:

$$Cost(K) = H_r^3 K^{\alpha+\beta} \cdot \frac{\log(1/\delta)}{\varepsilon^2} \cdot \left( \frac{1}{1 - \frac{1}{H_r} - \frac{1}{K^d}} \right)^2$$

To find the minimum, we differentiate with respect to $K$. Since the prefactor $H_r^3 \frac{\log(1/\delta)}{\varepsilon^2}$ is independent of $K$, we focus on $g(K) = K^{\alpha+\beta} \cdot \left( 1 - \frac{1}{H_r} - \frac{1}{K^d} \right)^2$. To simplify the calculation, we use logarithmic differentiation $\ln g(K) = (\alpha + \beta) \ln K + 2 \ln \left( 1 - \frac{1}{H_r} - \frac{1}{K^d} \right)$.

Differentiating both sides with respect to $K$: $\frac{g'(K)}{g(K)} = \frac{\alpha+\beta}{K} - \frac{2d}{K^{d+1}\left(1 - \frac{1}{H_r} - \frac{1}{K^d}\right)}$. At the optimal point $K^*$, we have $g'(K^*) = 0$, which gives:

$$\alpha + \beta = \frac{2d + \alpha + \beta}{(K^*)^d \left( 1 - \frac{1}{H_r} \right)}$$

Then $(K^*)^d = \frac{\alpha+\beta+2d}{(\alpha+\beta)\left(1 - \frac{1}{H_r}\right)}$. Taking the $d$-th root $K^* = \left( \frac{\alpha+\beta+2d}{(\alpha+\beta)\left(1 - H_r^{-1}\right)} \right)^{1/d}$.

We now analyze the expression for $K^*$ in the limit where $H_r \gtrsim \lambda_1 \gg 1$. This implies that $x = H_r^{-1}$ is a small positive value. We can rewrite the expression for $K^*$ as:

$$K^* = \left( \frac{2d + \alpha + \beta}{\alpha + \beta} \right)^{1/d} \cdot \left( 1 - H_r^{-1} \right)^{-1/d}$$

We focus on the second term, $(1 - H_r^{-1})^{-1/d}$. Using the generalized binomial approximation, $(1 - x)^a \approx 1 - ax$ for small $x$, where $x = H_r^{-1}$ and $a = -1/d$: $\left( 1 - H_r^{-1} \right)^{-1/d} \approx 1 + \frac{1}{dH_r}$. For large $\lambda_1$, the argument of the exponential is small. Using the first-order Taylor approximation $e^y \approx 1 + y$ for small $y$: $\left( 1 - H_r^{-1} \right)^{-1/d} \approx \exp\left( \frac{1}{d\lambda_1} \right)$. Substituting this back gives the final approximate form for the optimal resolution:

$$K^* \approx \left( \frac{2d + \alpha + \beta}{\alpha + \beta} \right)^{1/d} \cdot \exp\left( \frac{1}{d\lambda_1} \right)$$

This completes the proof.

□

## C.2 TOKAMAK PLASMA CONTROL

### C.2.1 ENVIRONMENT DESCRIPTION

In this task, we study a magnetically confined plasma system involving the control of a droplet-like plasma structure within a tokamak device. The objective is to manipulate the external magnetic field to maintain stable confinement of the plasma droplet and prevent wall contact (Degrave et al., 2022).

The governing dynamics are modeled by a coupled magnetohydrodynamic system with deformable interface dynamics, comprising the following equations:

$$\frac{\partial \rho}{\partial t} + \nabla \cdot (\rho \mathbf{v}) = 0 \qquad \text{(Continuity equation)}$$

$$\rho \left( \frac{\partial \mathbf{v}}{\partial t} + \mathbf{v} \cdot \nabla \mathbf{v} \right) = -\nabla p + \mathbf{J} \times \mathbf{B} + \mu \nabla^2 \mathbf{v} + \mathbf{F}_{\text{surface}} \qquad \text{(Momentum equation)}$$

$$\frac{\partial \mathbf{B}}{\partial t} = \nabla \times (\mathbf{v} \times \mathbf{B}) + \eta \nabla^2 \mathbf{B} \qquad \text{(Magnetic field control equation)}$$

$$\frac{\partial \phi}{\partial t} + \mathbf{v} \cdot \nabla \phi = 0 \qquad \text{(Deformation of the droplet's interface)}$$

$$\sigma \kappa = \gamma_{\text{surface}} \qquad \text{(Boundary conditions)}$$

Here, $\rho$ is the plasma density, $\mathbf{v}$ is the velocity field, $p$ is the pressure, $\mathbf{B}$ is the magnetic field, and $\mathbf{J} = \nabla \times \mathbf{B}$ is the current density. The term $\mu$ is the dynamic viscosity, $\eta$ is the magnetic resistivity, and $\mathbf{F}_{\text{surface}}$ represents interface forces. The function $\phi$ is a level-set representation of the droplet interface, and its advection governs shape deformation. The final equation enforces curvature-related surface force balance, where $\sigma$ is the surface tension coefficient, $\kappa$ is the local curvature, and $\gamma_{\text{surface}}$ represents surface forcing.

The spatial domain is discretized using a structured mesh aligned with the tokamak cross-section. A schematic of the cross-sectional mesh layout is shown in Figure C.2. The state space includes grid-sampled values of $\rho, \mathbf{v}, \mathbf{B}, \phi$, and other derived quantities such as vorticity and magnetic energy. Control is applied via a discrete set of actions adjusting time-varying coil currents $\mathbf{I}_{\text{coil}}(t)$, which shape the confinement field. At each decision step, the reinforcement learning agent selects a control signal that reshapes the magnetic confinement field.

The reward function is defined as the negative of the minimum distance between the plasma droplet and the tokamak boundary, promoting stable suspension and avoiding wall contact. The long-term objective is to maintain the droplet within the desired confinement region while suppressing shape instabilities and maximizing equilibrium duration.

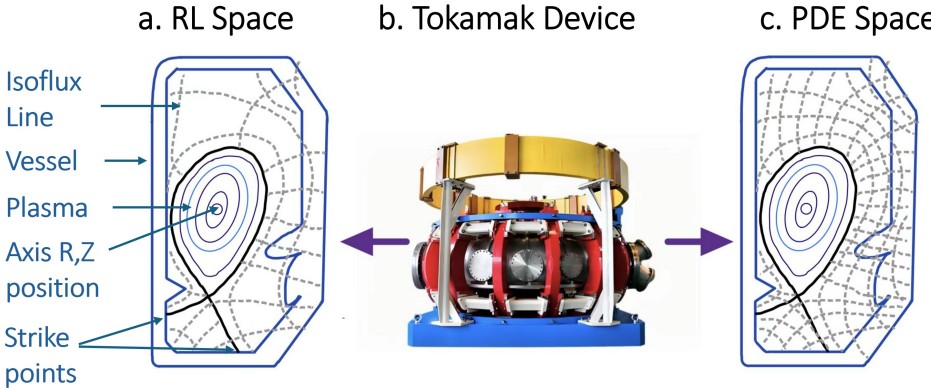

Figure C.2: Structured mesh layout of the tokamak poloidal cross-section.

### C.2.2 Error Coupling Between RL and PDE Spaces in AI4S Systems

In this section, we analyze the sources and scales of error introduced in AI4S systems, especially the bidirectional projection between the RL space and the PDE-based physical space. As discussed in the main text and lemma S.3, for systems with observation noise, achieving an unbiased estimation of the RL transition kernel imposes a lower bound on the temporal resolution of the RL system, which is determined by the intrinsic dynamics of the underlying physical process. Based on this property, we couple the errors in the RL space and the physical space, and ultimately derive an expression for the total system error. Then through a $\rho - K$ analysis, we reveals the key factors that govern the overall numerical error of the AI4S system.

We begin by introducing key notations to distinguish between two discretization structures:

The **RL space** is characterized by $(H_r, S_r, A_r)$, denoting the RL decision horizon, state discretization, and action granularity. The **physical (PDE) space** is represented by $(H_w, S_w, A_w)$, corresponding to the physical evolution horizon, spatial resolution, and control parameter resolution in the underlying PDE model.

Without loss of generality, we assume that the spatial grid resolution satisfies $\Delta x = \Delta y$. Accordingly, we define $\Delta \mathbf{x}_{p,\text{int}}$ as the spatial grid spacing inside the PDE-based surrogate environment, and $\Delta \mathbf{x}_{p,\text{bd}}$ as the spatial spacing along the domain boundary. Similarly, $\Delta \mathbf{x}_{r,\text{int}}$ and $\Delta \mathbf{x}_{r,\text{bd}}$ denote the interior and boundary grid resolutions within the RL environment, respectively.

Let the reinforcement learning policy operate on a coarse temporal scale, issuing actions every $\Delta t_r$ seconds, while the physical system evolves at a much finer resolution $\Delta t_w \ll \Delta t_r$. Consequently, each RL control interval consists of $N_t = \Delta t_r / \Delta t_w$ internal PDE integration steps. The total prediction error associated with a single RL-AI4S step is then the cumulative result of local errors incurred at each fine-grained PDE time step.

We now analyze the composition of prediction error in AI4S systems. During each RL interaction cycle, the observed state must be projected from the RL space to the PDE space, evolved forward under the PDE dynamics, and then projected back into the RL space. This bi-directional mapping introduces two distinct projection errors, both of which may be amplified near domain boundaries due to the nonlinear characteristics of the governing equations.

Therefore, the total error consists of three main components: discretization error intrinsic to the RL space, numerical integration error accumulated within the PDE solver and coupling error caused by the nontrivial projections between RL and PDE spaces.

**Fine-Grained Surrogate Error in PDE-Based Surrogate Environments.** We now analyze the surrogate error incurred at each fine-grained PDE step within the AI4S framework. This local error, denoted as $\Delta_{\phi}^{(k)}$, arises from three primary sources: discretization in the spatial domain, imprecision in the control action, and interpolation near the physical boundary.

We consider the level-set PDE that governs the evolution of an interface function $\phi$: $\frac{\partial \phi}{\partial t} + \mathbf{v} \cdot \nabla \phi = 0$, where $\phi = \phi(x,t)$ is the level-set function and $\mathbf{v} = \mathbf{v}(x,t)$ is the velocity field. This PDE is discretized in time using a forward Euler method and in space using central differences.

The numerical scheme used to evolve $\phi$ at a single time step is: $\frac{\phi_i^{n+1} - \phi_i^n}{\Delta t_w} + \mathbf{v}_i \cdot \nabla_h \phi_i^n = 0$, where $\nabla_h \phi_i^n$ is the discrete approximation of the gradient at grid point $x_i$ and time $t_n$, computed using central differences.

We now analyze the error introduced by this scheme, which is defined as the residual obtained by substituting the exact solution $\phi(x,t)$ into the numerical scheme.

**Temporal truncation error:** Applying the Taylor expansion of $\phi(x,t)$ in time at point $x_i$ gives:

$$\phi_i^{n+1} = \phi_i^n + \Delta t_w \left. \frac{\partial \phi}{\partial t} \right|_i^n + \frac{\Delta t_w^2}{2} \left. \frac{\partial^2 \phi}{\partial t^2} \right|_i^n + \mathcal{O}(\Delta t_w^3).$$

The finite difference approximation of the time derivative becomes:

$$\frac{\phi_i^{n+1} - \phi_i^n}{\Delta t_w} = \left.\frac{\partial \phi}{\partial t}\right|_i^n + \frac{\Delta t_w}{2}\left.\frac{\partial^2 \phi}{\partial t^2}\right|_i^n + \mathcal{O}(\Delta t_w^2).$$

To establish the temporal truncation error, we consider the second-order Taylor expansion of the level-set function $\phi(x,t)$ in time at a fixed spatial point $x_i$. In accordance with the physical model of tokamak plasma dynamics, the underlying equations exhibit parabolic-type behavior with smooth forcing and magnetic confinement fields. As such, we assume the solution $\phi(x,t)$ possesses sufficient temporal regularity.

In particular, we assume $\phi \in C^2([0,T] \times \Omega)$, which ensures that the second-order time derivative is uniformly bounded. That is, there exists a constant $C > 0$ such that: $\left|\frac{\partial^2 \phi}{\partial t^2}(x,t)\right| \leq C, \quad \forall (x,t) \in \Omega_T$. Under this regularity assumption, the finite difference approximation of the time derivative yields a local truncation error of the form: $\tau_{\text{time}} = \mathcal{O}(\Delta t_w)$, where the hidden constant depends on the supremum of $\left|\partial^2\phi/\partial t^2\right|$.

**Spatial truncation error:** We consider the approximation of the spatial gradient $\nabla \phi$ using central differences on a uniform grid with spacing $\Delta x = \Delta y = h$. For a function $\phi \in H^1(\Omega)$, we focus on the discrete gradient approximation defined as: $\left.\frac{\partial_h \phi}{\partial x}\right|_i := \frac{\phi(x_{i+1}) - \phi(x_{i-1})}{2h}$.

To evaluate the truncation error, we apply the first-order Taylor expansion $\phi(x_{i\pm 1}) = \phi(x_i) \pm h\phi'(x_i) + \mathcal{O}(h^2)$. Substituting into the difference formula yields $\frac{\phi(x_{i+1}) - \phi(x_{i-1})}{2h} = \phi'(x_i) + \mathcal{O}(h)$, which implies $\left.\frac{\partial_h \phi}{\partial x}\right|_i = \left.\frac{\partial \phi}{\partial x}\right|_i + \mathcal{O}(h)$. A similar result holds for the $y$-component. Combining both components, we obtain the discrete gradient $\nabla_h \phi = \nabla \phi + \mathcal{O}(h)$, which holds pointwise under the assumed regularity.

In the context of PDE-based surrogate environments, the spatial truncation error depends on the local grid region under consideration. Specifically, if we focus on the interior of the domain where the solution is sufficiently regular, the central difference approximation yields: $\nabla_h \phi = \nabla \phi + \mathcal{O}(\|\Delta \mathbf{x}_{p,\text{int}}\|)$. However, near the boundary or interface, the regularity of $\phi$ is reduced due to physical or geometric discontinuities. In such cases, the interpolation error is governed by the trace theorem, and satisfies only: $\|\phi - \phi_h\|_{L^2(\partial\Omega)} = \mathcal{O}(\|\Delta \mathbf{x}_{p,\text{bd}}\|^{1/2})$. For further details on the trace theorem and its implications for boundary error analysis, see Appendix D.

**Action-induced control error:** In our system, the velocity field $\mathbf{v}$, which governs the evolution of the plasma interface, is not directly actuated but indirectly induced through a cascade of physical interactions governed by the magnetohydrodynamic (MHD) equations. Specifically, the control input $a_w$ corresponds to the current applied to external boundary coils. The discretization of the action space introduces finite-resolution perturbations $\Delta a_w$, which in turn cause quantization-induced errors in the boundary actuation.

The control-to-interface influence chain is as follows:

$$\Delta a_w \to \mathbf{B}|_{\partial\Omega} \to \mathbf{J} = \nabla \times \mathbf{B} \to \mathbf{v} \to \phi.$$

The applied control signal perturbs the boundary magnetic field $\mathbf{B}|_{\partial\Omega}$ through electromagnetic models such as the Biot–Savart law or magnetic vector potentials $\delta\mathbf{B}|_{\partial\Omega} = \mathcal{O}(\Delta a_w)$. The magnetic field $\mathbf{B}$ evolves under the time-dependent induction equation $\frac{\partial \mathbf{B}}{\partial t} = \nabla \times (\mathbf{v} \times \mathbf{B}) + \eta\nabla^2\mathbf{B}$. This is a nonlinear parabolic PDE, where the diffusion term $\eta\nabla^2\mathbf{B}$ dominates the short-time behavior. Over a single control step, we can approximate the system as quasi-static and neglect the time derivative, resulting in $-\eta\nabla^2\mathbf{B} \approx \nabla \times (\mathbf{v} \times \mathbf{B})$. Under this elliptic approximation, boundary disturbances introduced by $\Delta a_w$ propagate into the interior via the smoothing effect of the Laplacian. By elliptic regularity and the trace theorem, we obtain: $\delta\mathbf{B}|_\Omega = \mathcal{O}(\Delta a_w^{1/2})$.

The momentum dynamics are governed by the MHD Navier–Stokes equation:

$$\rho\left(\frac{\partial \mathbf{v}}{\partial t} + \mathbf{v} \cdot \nabla\mathbf{v}\right) = -\nabla p + \mathbf{J} \times \mathbf{B} + \mu\nabla^2\mathbf{v} + \mathbf{F}_{\text{surface}},$$

where the Lorentz force $\mathbf{J} \times \mathbf{B}$ is the dominant actuator of plasma motion. Since $\mathbf{J} = \nabla \times \mathbf{B}$, both $\delta \mathbf{B}$ and $\delta \mathbf{J}$ scale as $\mathcal{O}(\Delta a_w^{1/2})$. This yields a force perturbation:

$$\delta \mathbf{F}_{\text{Lorentz}} = \delta \mathbf{J} \times \mathbf{B} + \mathbf{J} \times \delta \mathbf{B} = \mathcal{O}(\Delta a_w^{1/2}),$$

and correspondingly a sublinear response in velocity: $\Delta \mathbf{v} = \mathcal{O}(\Delta a_w^{1/2})$.

**Final per-step surrogate error:** We now summarize the local surrogate error incurred in the discretized interface evolution governed by: $\frac{\partial \phi}{\partial t} + \mathbf{v} \cdot \nabla \phi = 0$. Each term introduces a distinct approximation error: The time derivative $\partial_t \phi$ is discretized by forward Euler, incurring $\mathcal{O}(\Delta t_w)$ truncation error. The spatial gradient $\nabla \phi$ is computed via finite differences, contributing $\mathcal{O}(\|\Delta \mathbf{x}_{p,\text{int}}\|)$ in the interior and $\mathcal{O}(\|\Delta \mathbf{x}_{p,\text{bd}}\|^{1/2})$ near boundaries. The velocity field $\mathbf{v}$ is perturbed by action discretization, with induced error $\mathcal{O}(\Delta a_w^{1/2})$.

Substituting these into the PDE gives:

$$\frac{\phi^{n+1} - \phi^n}{\Delta t_w} + (\mathbf{v} + \mathcal{O}(\Delta a_w^{1/2})) \cdot \left( \nabla \phi + \mathcal{O}(\|\Delta \mathbf{x}_{p,\text{bd}}\|^{1/2}) \right),$$

yielding the total local surrogate error per step as:

$$\Delta_\phi^{(k)} = \mathcal{O}(\Delta t_w) + \mathcal{O}(\|\Delta \mathbf{x}_{p,\text{int}}\|) + \mathcal{O}(\Delta a_w^{1/2}) + \mathcal{O}(\|\Delta \mathbf{x}_{p,\text{bd}}\|^{1/2}) + \mathcal{O}(\Delta a_w^{1/2} \cdot \|\Delta \mathbf{x}_{p,\text{bd}}\|^{1/2}).$$

**One-Step Prediction Error in RL-Based Control Environments.** In the AI4S control framework, the RL agent interacts with a PDE-based simulator by issuing a discretized action $a_r$ based on an observed state $s_r$, which itself is a downsampled or filtered version of the true PDE state $y \in \mathcal{Y}$. The PDE system then evolves the dynamics over a time horizon $\Delta t_r$ and returns a new state $s_r'$. The surrogate prediction error in this interaction arises from discretization of state and action inputs and integration-induced temporal accumulation.

**RL observation error as truncation error in PDE-Based environments:** In PDE-based reinforcement learning environments, the agent does not observe the full continuous physical state $y(x) \in \mathbb{R}^d$, but rather a discretized version $s_r \in \mathbb{R}^d$, obtained via sampling or interpolation over a coarse spatial mesh. This introduces a spatial offset between the actual physical location $x$ and the evaluation point $\tilde{x}$ used in downstream PDE computation: $\|x - \tilde{x}\| \leq \|\Delta \mathbf{x}_{r,\text{int}}\|$. Such an offset mimics the numerical behavior of spatial truncation errors in classical finite difference schemes.

Consider a PDE used for control computation in the simulator: $\frac{\partial \phi}{\partial t} + \mathbf{v} \cdot \nabla \phi = 0$, the derivative $\nabla \phi$ is evaluated not at the true point $x$, but at $\tilde{x}$. The following analysis parallels our earlier fine-grained surrogate error in PDE-based surrogate environments, from which we conclude that the truncation error induced by RL's observation of the state scales as $\mathcal{O}(\|\Delta \mathbf{x}_{r,\text{int}}\|)$.

Similarly, near the domain boundary $\partial \Omega$, the solution $\phi \in H^1(\Omega)$ typically lacks full smoothness. By the trace theorem, the restriction of $\phi$ to the boundary lies in the fractional Sobolev space: $\phi|_{\partial \Omega} \in H^{1/2}(\partial \Omega)$. Consequently, interpolation errors on the boundary are governed by lower regularity, yielding $\|\phi - \phi_h\|_{L^2(\partial \Omega)} = \mathcal{O}(\|\Delta \mathbf{x}_{r,\text{bd}}\|^{1/2})$, which mirrors the $\mathcal{O}(h^{1/2})$ rate of convergence found in boundary discretization analysis of finite difference schemes.

In summary, the state observation errors in RL, when incorporated into the PDE solver, act as implicit spatial discretization errors. Their scaling matches classical truncation theory:

$$\Delta_{\text{state}} = \begin{cases} \mathcal{O}(\|\Delta \mathbf{x}_{r,\text{int}}\|), & \text{for interior evaluation,} \\ \mathcal{O}(\|\Delta \mathbf{x}_{r,\text{bd}}\|^{1/2}), & \text{for boundary evaluation.} \end{cases}$$

**Action discretization error:** The agent selects an action $a_r$ from a discretized control space, which governs the boundary actuation of the PDE system. The true control variable $a$ is approximated by $a_r$, with finite resolution $\Delta a_r$. This discretization introduces quantization error in the applied boundary signal.

Similar to our earlier fine-grained surrogate error analysis in PDE-based environments, we observe that this control mismatch propagates through the MHD system in a nonlinear and spatially diffused manner. The resulting perturbation in the velocity field—responsible for evolving the system

state—scales sublinearly with respect to the action resolution: $\Delta a_r \Rightarrow \delta \mathbf{v} = \mathcal{O}(\Delta a_r^{1/2})$. This velocity perturbation directly enters the PDE dynamics and contributes to the overall one-step prediction error in the RL loop.

**Temporal integration error:** During each RL step of duration $\Delta t_r$, the PDE simulator executes $N_t = \Delta t_r / \Delta t_w$ fine-grained steps with timestep $\Delta t_w$. At each substep, the forward Euler integration induces local truncation error: $\tau_t^{(n)} = \mathcal{O}(\Delta t_w)$, so that the cumulative integration error over one RL step becomes: $\sum_{n=1}^{N_t} \tau_t^{(n)} = \mathcal{O}(\Delta t_r)$.

Summing all contributions, the one-step surrogate prediction error in RL space satisfies:

$$\Delta_{\mathrm{RL}} = \underbrace{C_1 \|\Delta \mathbf{x}_{r,\mathrm{int}}\|}_{\text{State (interior)}} + \underbrace{C_2 \|\Delta \mathbf{x}_{r,\mathrm{bd}}\|^{1/2}}_{\text{State (boundary)}} + \underbrace{C_3 \Delta a_r^{1/2}}_{\text{Action resolution}} + \underbrace{C_4 \Delta t_r}_{\text{Temporal propagation}} .$$

**Refined Total Error Decomposition with Time-Scale Separation.** Based on our previous analysis of surrogate error in both the PDE and RL components, we now integrate all sources of discretization and numerical error into a unified framework. Over a single RL step, the total prediction error can be decomposed as follows:

$$\Delta_{\mathrm{total}} = \underbrace{C_1 \|\Delta \mathbf{x}_{r,\mathrm{int}}\|}_{\text{RL (interior space)}} + \underbrace{C_2 \|\Delta \mathbf{x}_{r,\mathrm{bd}}\|^{1/2}}_{\text{RL (boundary space)}} + \underbrace{C_3 \Delta a_r^{1/2}}_{\text{RL (action space)}} + \underbrace{C_4 \Delta t_r}_{\text{RL (time)}}$$
$$+ \underbrace{C_5 \frac{\Delta t_r}{\Delta t_w} \left( \|\Delta \mathbf{x}_{p,\mathrm{int}}\| + \Delta a_w^{1/2} + \|\Delta \mathbf{x}_{p,\mathrm{bd}}\|^{1/2} + \Delta a_w^{1/2} \cdot \|\Delta \mathbf{x}_{p,\mathrm{bd}}\|^{1/2} \right)}_{\text{PDE surrogate error (space + action + boundary), amplified by time scale separation}} . \tag{C.6}$$

To ensure that the learned RL transition kernel remains robust under observation uncertainty $\Delta y$, we require that: $\Delta_{\mathrm{total}} = \mathcal{O}(\Delta y)$. This constraint implies a resolution matching condition across all discretization dimensions. Specifically, we obtain the following asymptotic scaling relations:

$$\|\Delta \mathbf{x}_{r,\mathrm{int}}\| \sim \|\Delta \mathbf{x}_{p,\mathrm{int}}\| \sim \Delta a_r^{1/2} \sim \Delta a_w^{1/2} \sim \Delta t_r \sim \Delta y,$$
$$\|\Delta \mathbf{x}_{r,\mathrm{bd}}\| \sim \|\Delta \mathbf{x}_{p,\mathrm{bd}}\| \sim \Delta y^2, \quad \Delta t_w \ll \Delta y. \tag{C.7}$$

$\rho$-$K$ **Analysis.** We now turn to the analysis of the error rate associated with predicting the next state in AI4S environments based on an RL agent's current observation and action. Specifically, we define a prediction error event as one where the next predicted state fails to fall within the correct discretized RL state cell. Accordingly, we define the relative prediction error rate as: $\rho = 1 - \frac{\Delta y}{\text{Total Prediction Error}}$, where $\Delta y$ is the size of the RL spatial grid. In a three-dimensional state space, it corresponds to the cube of the grid length.

The total prediction error in the system arises from a combination of three interrelated sources. First, observation noise introduces inherent uncertainty, stemming from either partial observability or sensor imprecision within the reinforcement learning environment. Second, numerical surrogate error which quantified earlier in Equation equation C.6 captures the discretization-induced inaccuracies originating from both the RL and PDE components. Finally, the intrinsic growth of initial perturbations, as detailed in Lemma S.3, contributes an additional error term due to the modal amplification behavior of the MHD system; specifically, this leads to a forward-propagated uncertainty of order $\lambda_1 \Delta y / H_r$, where $\lambda_1$ denotes the leading eigenvalue and $H_r$ is the RL planning horizon.

Now we further quantify the numerical surrogate error based on the resolution matching relations previously derived. Given the grid refinement ratio $K := \Delta \mathbf{x}_r / \Delta \mathbf{x}_p$, which compares the spatial resolution of the PDE environment to that of the RL environment, and considering the combined error impact across three dimensions in state space, we obtain: $\Delta_{\mathrm{total}} \sim \frac{C_1}{K^3} \Delta y^{(j)}$. Substituting all three error components into the definition of $\rho$, we obtain:

$$\rho = 1 - \frac{\Delta y}{\lambda_1 \Delta y / H_r + \Delta y + \Delta_{\mathrm{total}}} = 1 - \frac{1}{\lambda_1 / H_r + 1 + C_1 / K^3}.$$

So in the limit of high-resolution settings, i.e., when $H_r$ and $K$ are large, we obtain $\rho = \mathcal{O}\left(\frac{1}{H_r} + \frac{1}{K^3}\right)$.

### C.2.3    Optimal Computational Cost Allocation between RL and AI4S

In this section, building on the previous analysis of error scaling, we provide a detailed explanation of how to align the discretization parameters of reinforcement learning and AI4S models to achieve optimal computational efficiency.

It is evident that higher fidelity in the physical model leads to more accurate predictions, which in turn accelerates convergence to the optimal policy in RL. However, since AI4S simulators are typically executed under fixed computational budgets, it is crucial to ensure that the computational load in the RL loop is of the same order as that of the AI4S module. Our objective is to minimize total computational cost by optimally distributing resolution between the two systems.

According to Lemma B.1, the sample complexity or computational cost of the RL component can be expressed as $\text{Cost}_{\text{RL}} = \mathcal{O}\left(\frac{H_r^3 S_r A_r \cdot \log(1/\delta)}{\varepsilon^2}\right)$. Meanwhile, the computational cost of the AI4S physical simulator is governed by its spatiotemporal resolution, and can be approximated by: $\text{Cost}_{\text{AI4S}} = \mathcal{O}(H_w S_w A_w)$.

To ensure computational consistency, we aim to align these quantities. This alignment is further guided by the theoretical result in Lemma S.3, where the RL horizon $H_r$ is constrained by the intrinsic dynamics of the PDE system: $H_r \lesssim \frac{1}{\lambda_1}$, where $\lambda_1$ is the dominant growth rate in the linearized MHD system.

Based on the resolution matching conditions derived earlier, we assume the following scalings:

$$A_r = \mathcal{O}(A_w) = \mathcal{O}(x_w^2),\, S_r = \mathcal{O}(S_w) = \mathcal{O}(x_w^6),\, H_r = H_w^{1/3} \cdot \mathcal{O}(x_w).$$

Here, $x_w$ denotes the effective resolution of the AI4S spatial grid. The scaling $S_r = \mathcal{O}(x_w^6)$ arises from both the intrinsic dimensionality of the state space and the additional resolution requirements near the domain boundary. In tokamak plasma systems, the physical state space is inherently three-dimensional. Under uniform discretization, this implies that the number of internal state variables scales as $\mathcal{O}(x_w^3)$.

Besides, near the boundary $\partial\Omega$, the PDE solution often exhibits reduced regularity—formally, if $y \in H^1(\Omega)$, then its restriction to the boundary lies in the fractional Sobolev space $H^{1/2}(\partial\Omega)$, as stated by the trace theorem. This lower regularity implies that to maintain the same level of approximation accuracy at the boundary, the spatial grid must be refined further. In particular, the resolution must be squared in order to compensate for the smoothness loss, effectively contributing an additional factor of $\mathcal{O}(x_w^3)$ from the boundary region.

The total RL cost then becomes:

$$\text{Cost}_{\text{RL}} \propto \frac{H_r S_r A_r \cdot \log(1/\delta)}{(\Delta y^{(j)} - p_{\max}^{(j)})^2} \approx \frac{H_r^9 K^8}{\varepsilon} \cdot \left(\frac{1}{1 - \frac{1}{H_r} - \frac{1}{K^3}}\right)^2, \tag{C.8}$$

Our goal is to find the optimal refinement ratio $K^*$ that minimizes this cost expression. Formally, we solve the following optimization problem:

$$\min_{K>0} \quad C(K) := \frac{H_r^9 K^8}{\varepsilon} \cdot \left(\frac{1}{1 - \frac{1}{H_r} - \frac{1}{K^3}}\right)^2.$$

To find the stationary point, we differentiate $C(K)$ with respect to $K$ and set the derivative to zero: $\frac{dC}{dK} = 0$. Denote $\alpha := 1 - \frac{1}{H_r}$, then $C(K) \propto K^8 \left(\frac{1}{\alpha - \frac{1}{K^3}}\right)^2$. Taking logarithmic derivative: $\frac{d \log C}{dK} = \frac{8}{K} - 2 \cdot \frac{3}{K^4(\alpha - \frac{1}{K^3})}$. Setting this derivative to zero and solving yields the optimal $K^*$: $K^* = \left(\frac{7}{4(1 - \frac{1}{H_r})}\right)^{1/3}$. So in the high-resolution regime where $H_r \gg 1$, we approximate:

$$K^* \gtrsim \left(\frac{7}{4(1 - \frac{1}{\lambda_1})}\right)^{1/3} \approx \left(\frac{7}{4} e^{1/\lambda_1}\right)^{1/3}.$$

### C.3 TURBULENT AIRFOIL REGULATION

#### C.3.1 ENVIRONMENT DESCRIPTION

In this task, we study an unsteady aerodynamic system involving the control of a turbulent airfoil flow field. The objective is to regulate the pitching angle of the airfoil to manipulate the surrounding flow structure and ultimately maximize lift (Portal-Porras et al., 2022).

The governing equations for the fluid dynamics are the two-dimensional incompressible Navier-Stokes equations:

$$\frac{\partial \mathbf{u}}{\partial t} + (\mathbf{u} \cdot \nabla)\mathbf{u} = -\nabla p + \nu \Delta \mathbf{u}$$

where $\mathbf{u}(x, y, t) = (u(x, y, t), v(x, y, t))$ is the velocity vector field, $p(x, y, t)$ denotes the pressure, $\nu$ is the kinematic viscosity, $\nabla$ is the gradient operator, and $\Delta$ is the Laplacian operator.

The system is also subject to the incompressibility constraint: $\nabla \cdot \mathbf{u} = 0$. These equations define the error sources and discretization structure that form the basis for our multi-resolution analysis.

In this environment, we adopt a structured grid discretization for the spatial domain, employing an O-grid topology that conforms to the geometry of the airfoil. As illustrated in Figure C.3.a, the mesh is progressively refined near the surface of the airfoil to accurately resolve the boundary layer. The state space is thus defined over this discretized grid . The action space is discretized into a finite set of pitch angle increments. At each time step, the RL agent selects a discrete action corresponding to a rotation of the airfoil about its chord line. The reward function is constructed based on the instantaneous lift coefficient, with the overall objective of maximizing its long-term average.

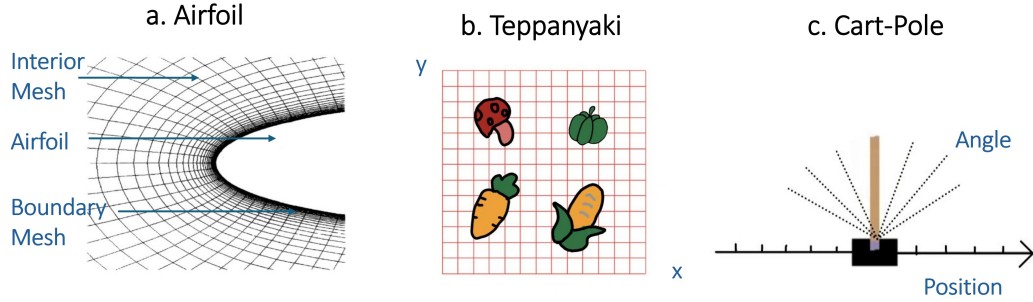

Figure C.3: Visualization of Discretization Structures in Different Control Environments.

#### C.3.2 ERROR COUPLING BETWEEN RL AND PDE SPACES IN AI4S SYSTEMS

Since the state space of the turbulent airfoil flow field also features a division between interior and boundary regions, we adopt an analysis framework analogous to that used for the tokamak device to analyze the composition of prediction error in AI4S systems.

**Fine-Grained Surrogate Error in Airfoil Flow Environments.** We now analyze the surrogate error incurred at each fine-grained PDE step within AI4S framework. In airfoil flow control tasks, the dynamics are governed by the two-dimensional incompressible Navier–Stokes equations: $\frac{\partial \mathbf{u}}{\partial t} + (\mathbf{u} \cdot \nabla)\mathbf{u} = -\nabla p + \nu \Delta \mathbf{u}$. This PDE is discretized in time using a forward Euler method and in space using central differences.

**Temporal truncation error:** Using the Taylor expansion at time $t_n$, we write:

$$\mathbf{u}^{n+1} = \mathbf{u}^n + \Delta t_w \left.\frac{\partial \mathbf{u}}{\partial t}\right|^n + \frac{1}{2}\Delta t_w^2 \left.\frac{\partial^2 \mathbf{u}}{\partial t^2}\right|^n + \mathcal{O}(\Delta t_w^3).$$

Then the numerical derivative satisfies:

$$\frac{\mathbf{u}^{n+1} - \mathbf{u}^n}{\Delta t_w} = \left.\frac{\partial \mathbf{u}}{\partial t}\right|^n + \frac{1}{2}\Delta t_w \left.\frac{\partial^2 \mathbf{u}}{\partial t^2}\right|^n + \mathcal{O}(\Delta t_w^2),$$

which implies a local truncation error of $\mathcal{O}(\Delta t_w)$ provided $\mathbf{u} \in C^2$.

**Spatial truncation error:** For a function $u \in H^1(\Omega)$, we focus on the numerical approximation of the convective term $(\mathbf{u} \cdot \nabla)\mathbf{u}$ and the diffusive term $\nu\Delta\mathbf{u}$, which appear in the Navier-Stokes momentum equation: $\frac{\partial \mathbf{u}}{\partial t} + (\mathbf{u} \cdot \nabla)\mathbf{u} = -\nabla p + \nu\Delta\mathbf{u}$. For the convective term, take one component $u(x,t)$, and use central differences to approximate: $u\frac{\partial u}{\partial x}\big|_{x_i} \approx u(x_i) \cdot \frac{u(x_{i+1})-u(x_{i-1})}{2h}$. Since the function $u \in H^1(\Omega)$, the accuracy of the Taylor expansion is limited to $\mathcal{O}(h^2)$. Applying Taylor expansion, we obtain: $\frac{u(x_{i+1})-u(x_{i-1})}{2h} = u'(x_i) + \mathcal{O}(h)$, which leads to: $u(x_i) \cdot \frac{u(x_{i+1})-u(x_{i-1})}{2h} = u(x_i)u'(x_i) + \mathcal{O}(h)$.

For the viscous diffusion term, the Laplacian $\Delta u$ is approximated using second-order central differences: $\frac{\partial^2 u}{\partial x^2}\big|_{x_i} \approx \frac{u(x_{i+1})-2u(x_i)+u(x_{i-1})}{h^2}$, and similarly in the $y$-direction. From Taylor expansion: $\frac{u(x_{i+1})-2u(x_i)+u(x_{i-1})}{h^2} = u''(x_i) + \mathcal{O}(h)$. Hence, the Laplacian approximation also introduces an error of $\mathcal{O}(h)$.

Therefore, both nonlinear convective and diffusive terms yield local discretization errors that can be bounded as:

$$(\mathbf{u} \cdot \nabla_h)\mathbf{u} = (\mathbf{u} \cdot \nabla)\mathbf{u} + \mathcal{O}(\|\Delta\mathbf{x}_{p,\text{int}}\|), \quad \Delta_h\mathbf{u} = \Delta\mathbf{u} + \mathcal{O}(\|\Delta\mathbf{x}_{p,\text{int}}\|).$$

On or near the airfoil surface $\partial\Omega$, the velocity field $\mathbf{u}$ may possess only $H^1$ regularity. By the trace theorem, its restriction belongs to: $\mathbf{u}|_{\partial\Omega} \in H^{1/2}(\partial\Omega)$, and the numerical interpolation error along the boundary grid satisfies: $\|\mathbf{u} - \mathbf{u}_h\|_{L^2(\partial\Omega)} = \mathcal{O}(\|\Delta\mathbf{x}_{p,\text{bd}}\|^{1/2})$.

**Action-induced control error:** In airfoil flow control, boundary actuation modifies wall velocities. Suppose this control is discretized with resolution $\Delta a_w$. The boundary perturbation in velocity is: $\delta\mathbf{u}|_{\partial\Omega} = \mathcal{O}(\Delta a_w)$. Because the viscous diffusion equation governs short-time dynamics: $\frac{\partial \mathbf{u}}{\partial t} = \nu\Delta\mathbf{u}$, we model the induced field internally by elliptic smoothing, yielding: $\delta\mathbf{u}|_\Omega = \mathcal{O}(\Delta a_w^{1/2})$.

**Final surrogate error per step:** Combining the above terms, the total local surrogate error per PDE step in the airfoil flow simulator is:

$$\Delta_\mathbf{u}^{(k)} = \mathcal{O}(\Delta t_w) + \mathcal{O}(\|\Delta\mathbf{x}_{p,\text{int}}\|) + \mathcal{O}(\Delta a_w^{1/2}) + \mathcal{O}(\|\Delta\mathbf{x}_{p,\text{bd}}\|^{1/2}).$$

**One-Step Prediction Error in RL-Controlled Airfoil Environments.** Based on our earlier surrogate error decomposition for tokamak systems, we now analyze the one-step prediction error in RL-based control of unsteady airfoil flow. The underlying dynamics are governed by the 2D incompressible Navier-Stokes equations: $\frac{\partial \mathbf{u}}{\partial t} + (\mathbf{u} \cdot \nabla)\mathbf{u} = -\nabla p + \nu\Delta\mathbf{u}$. In this setting, the RL agent selects an action $a_r$ based on an observed state $s_r$, which is a spatially discretized and possibly noisy version of the continuous flow field $\mathbf{u}$. The PDE simulator uses $a_r$ to modify boundary conditions, evolves the fluid state over a macro timestep $\Delta t_r$, and returns the next observed state $s'_r$.

**RL observation error as truncation error:** Due to spatial discretization, the observation $s_r$ deviates from the true field $\mathbf{u}(x)$ by at most the grid resolution: $\|x - \tilde{x}\| \leq \|\Delta\mathbf{x}_{r,\text{int}}\|$. This offset behaves analogously to a spatial truncation error in finite difference schemes. As previously analyzed, for smooth internal flow: $\Delta_{\text{state}}^{\text{int}} = \mathcal{O}(\|\Delta\mathbf{x}_{r,\text{int}}\|)$. Near the airfoil surface, where boundary layers reduce regularity, trace theorem implies: $\|\mathbf{u} - \mathbf{u}_h\|_{L^2(\partial\Omega)} = \mathcal{O}(\|\Delta\mathbf{x}_{r,\text{bd}}\|^{1/2})$.

**Action discretization error:** The discretized control signal $a_r$ influences boundary actuation. Finite resolution $\Delta a_r$ introduces quantization error, which affects boundary velocity: $\delta\mathbf{u}|_{\partial\Omega} = \mathcal{O}(\Delta a_r)$. Via the viscous smoothing governed by $\nu\Delta\mathbf{u}$, this error diffuses into the interior: $\delta\mathbf{u}|_\Omega = \mathcal{O}(\Delta a_r^{1/2})$.

**Temporal integration error:** Each RL step of length $\Delta t_r$ consists of $N_t = \Delta t_r/\Delta t_w$ fine-scale Euler updates with local error $\mathcal{O}(\Delta t_w)$. Total accumulation yields: $\sum_{n=1}^{N_t} \tau_t^{(n)} = \mathcal{O}(\Delta t_r)$.

**Final surrogate error:** Combining all contributions, we obtain the total RL one-step prediction error:

$$\Delta_{\text{RL}} = \underbrace{C_1 \|\Delta\mathbf{x}_{r,\text{int}}\|}_{\text{State (interior)}} + \underbrace{C_2 \|\Delta\mathbf{x}_{r,\text{bd}}\|^{1/2}}_{\text{State (boundary)}} + \underbrace{C_3 \Delta a_r^{1/2}}_{\text{Action resolution}} + \underbrace{C_4 \Delta t_r}_{\text{Temporal integration}} .$$

**Refined Total Error Decomposition with Time-Scale Separation.** Based on our prior analysis of surrogate errors in both PDE and RL components, we now present a unified decomposition of all numerical and discretization errors. Over a single RL step, the total prediction error can be expressed as:

$$\Delta_{\text{total}} = \underbrace{C_1 \|\Delta\mathbf{x}_{r,\text{int}}\|}_{\text{RL (interior space)}} + \underbrace{C_2 \|\Delta\mathbf{x}_{r,\text{bd}}\|^{1/2}}_{\text{RL (boundary space)}} + \underbrace{C_3 \Delta a_r^{1/2}}_{\text{RL (action space)}} + \underbrace{C_4 \Delta t_r}_{\text{RL (time)}}$$
$$+ \underbrace{C_5 \frac{\Delta t_r}{\Delta t_w} \left( \|\Delta\mathbf{x}_{p,\text{int}}\| + \Delta a_w^{1/2} + \|\Delta\mathbf{x}_{p,\text{bd}}\|^{1/2} \right)}_{\text{PDE surrogate error amplified by time-scale separation}}. \tag{C.9}$$

To maintain robustness of the RL transition kernel under observation uncertainty $\Delta y$, we impose the constraint: $\Delta_{\text{total}} = \mathcal{O}(\Delta y)$. This leads to the following resolution matching conditions across discretization components:

$$\|\Delta\mathbf{x}_{r,\text{int}}\| \sim \|\Delta\mathbf{x}_{p,\text{int}}\| \sim \Delta a_r^{1/2} \sim \Delta a_w^{1/2} \sim \Delta t_r \sim \Delta y,$$
$$\|\Delta\mathbf{x}_{r,\text{bd}}\| \sim \|\Delta\mathbf{x}_{p,\text{bd}}\| \sim \Delta y^2, \quad \Delta t_w \ll \Delta y. \tag{C.10}$$

$\rho$-$K$ **Analysis.** We now turn to the analysis of the error rate associated with predicting the next state in AI4S environments based on an RL agent's current observation and action. We define the relative prediction error rate as: $\rho = 1 - \frac{\Delta y}{\text{Total Prediction Error}}$, where $\Delta y$ is the size of the RL spatial grid. In a two-dimensional state space, it corresponds to the square of the grid length. Similar to the previous analysis, the total prediction error comprises three key components: observation noise, numerical surrogate error, and intrinsic growth of initial perturbation.

Now we further quantify the numerical surrogate error based on the resolution matching relations previously derived. Considering the combined error impact across two dimensions in state space, we obtain: $\Delta_{\text{total}} \sim \frac{C_1}{K^2} \Delta y^{(j)}$. Substituting all three error components into the definition of $\rho$, we obtain:

$$\rho = 1 - \frac{\Delta y}{\lambda_1 \Delta y / H_r + \Delta y + \Delta_{\text{total}}} = 1 - \frac{1}{\lambda_1 / H_r + 1 + C_1 / K^2}.$$

So in the limit of high-resolution settings, i.e., when $H_r$ and $K$ are large, we obtain: $\rho = \mathcal{O}\left(\frac{1}{H_r} + \frac{1}{K^2}\right)$. The subsequent results on optimal computational cost allocation are summarized in Table 1 of the main text. As this involves only straightforward calculations and scaling arguments, we omit detailed elaboration here.

## C.4 TEPPANYAKI HEAT SEQUENCING

### C.4.1 ENVIRONMENT DESCRIPTION

In this task, we study a two-dimensional heat transfer system representing a teppanyaki cooking surface. The goal is to use reinforcement learning to control the heat distribution across the iron plate to ensure the fastest and most uniform cooking of food.

The thermal evolution of the plate is governed by the two-dimensional heat diffusion equation:

$$\frac{\partial T(x,y,t)}{\partial t} = \alpha \left( \frac{\partial^2 T}{\partial x^2} + \frac{\partial^2 T}{\partial y^2} \right) + Q(x,y,t),$$

where $T(x,y,t)$ denotes the temperature at spatial position $(x,y)$ and time $t$; $\alpha$ is the thermal diffusivity coefficient, determined by the material of the iron plate; $Q(x,y,t)$ represents the external heat source, which can be actively controlled by the agent at each grid cell (Koehler et al., 2024).

The spatial domain is discretized using a uniform Cartesian grid. Each grid cell corresponds to a controllable heating unit, allowing the RL agent to adjust the local heat input. As shown in Figure C.3.b. The action space is discretized into finite heating levels for each control unit. The overall objective is to optimize heating strategies that ensure efficient thermal convergence within the shortest possible time.

### C.4.2 ERROR COUPLING BETWEEN RL AND PDE SPACES IN AI4S SYSTEMS

Unlike the previously introduced two environments, the teppanyaki setting adopts a uniform Cartesian grid discretization for the spatial domain. As a result, there is no distinction between interior and boundary regions, and no precision loss due to boundary regularity degradation. In the following, we adopt an analysis framework analogous to that used for the tokamak device to examine the composition of prediction error in AI4S systems.

**Fine-Grained Surrogate Error in Teppanyaki Heat Diffusion Environment.** We now analyze the local surrogate error incurred in each fine-grained PDE step under the teppanyaki heat diffusion setting. The governing equation for the temperature field $T(x, y, t)$ over the iron plate is the two-dimensional heat diffusion equation: $\frac{\partial T}{\partial t} = \alpha \Delta T + Q(x, y, t)$, where $\alpha$ is the thermal diffusivity and $Q$ is the external heating source, modulated by the control input.

**Temporal truncation error:** We discretize the temporal derivative using the forward Euler scheme: $\frac{T_{i,j}^{n+1} - T_{i,j}^n}{\Delta t_w} = \alpha \Delta_h T_{i,j}^n + Q_{i,j}^n$, where $\Delta_h T_{i,j}^n$ is the discrete Laplacian at grid point $i, j$. Applying Taylor expansion, the numerical derivative satisfies:

$$\frac{\mathbf{T}_{i,j}^{n+1} - \mathbf{T}_{i,j}^n}{\Delta t_w} = \left.\frac{\partial \mathbf{T}}{\partial t}\right|_{i,j}^n + \frac{1}{2}\Delta t_w \left.\frac{\partial^2 \mathbf{T}}{\partial t^2}\right|_{i,j}^n + \mathcal{O}(\Delta t_w^2),$$

which implies a local truncation error of $\mathcal{O}(\Delta t_w)$ provided $\mathbf{T} \in C^2$. We obtain the temporal truncation error:$\tau_{\text{time}} = \mathcal{O}(\Delta t_w)$.

**Spatial truncation error:** The Laplacian operator is approximated by central differences:

$$\Delta T \approx \frac{T_{i+1,j} - 2T_{i,j} + T_{i-1,j}}{(\Delta x)^2} + \frac{T_{i,j+1} - 2T_{i,j} + T_{i,j-1}}{(\Delta y)^2}.$$

Assuming $T \in H^1(\Omega)$, the second-order spatial derivatives exist in the weak sense, but due to limited regularity we expect that: $\Delta_h T = \Delta T + \mathcal{O}(\|\Delta \mathbf{x}_p\|)$. This yields a spatial truncation error: $\tau_{\text{space}} = \mathcal{O}(\|\Delta \mathbf{x}_p\|)$.

**Action-induced control error:** In this environment, control is implemented by adjusting the external heat source term $Q(x, y, t)$ in the heat diffusion equation. Let the continuous control signal be denoted by $a(t)$, and suppose it is discretized with resolution $\Delta a_w$ into a piecewise constant control $a_w(t)$. Assuming that the mapping from the control input to the heat source is linear , we obtain: $Q(x, y, t) = \bar{Q}(x, y, t) + \delta Q(x, y, t)$, where $\bar{Q}$ is the nominal source corresponding to $a_w(t)$, and the perturbation in the source is: $\delta Q(x, y, t) = f(\delta a) = \mathcal{O}(\Delta a_w)$.

Since the heat equation is linear in $Q$, the perturbation in $Q$ propagates linearly to the temperature field. Hence, the resulting temperature deviation induced by control discretization satisfies: $\delta T = \mathcal{O}(\Delta a_w)$.

**Total per-step surrogate error.** Combining all error sources, the total surrogate error in the temperature prediction per PDE step is: $\Delta_T^{(k)} = \mathcal{O}(\Delta t_w) + \mathcal{O}(\|\Delta \mathbf{x}_p\|) + \mathcal{O}(\Delta a_w)$.

**One-Step Prediction Error in RL-Controlled Heat Diffusion Environments.** In the teppanyaki plate task modeled by the heat diffusion equation, the RL agent interacts with a PDE simulator by issuing a discretized action $a_r$ and observing a discretized temperature field $s_r$, which is derived from the true temperature field $T(x, y, t)$ through uniform spatial sampling. The simulator then evolves the system over a time step $\Delta t_r$ and returns an updated observation $s_r'$. The one-step prediction error in this process arises from three main sources: state observation error, action discretization error, and temporal integration error.

**RL observation error:** The observed state $s_r$ is obtained by projecting the continuous temperature field $T(x, y, t)$ onto a coarsely discretized uniform Cartesian grid. This introduces a spatial observation error due to the offset between the true location $(x, y)$ and its discrete representation $\tilde{x}, \tilde{y}$.

Assuming $T \in H^1(\Omega)$, this interpolation or sampling procedure introduces first-order truncation error: $\|T(x, y) - T(\tilde{x}, \tilde{y})\| = \mathcal{O}(\|\Delta\mathbf{x}_{r,\text{int}}\|)$. This matches classical interpolation theory over uniform grids and reflects the interior spatial error due to limited resolution in RL's perception of the state.

**Action discretization error:** The action space is discretized with resolution $\Delta a_r$. The true continuous control $a$ is approximated by its discretized counterpart $a_r$, and thus the perturbation in the heat source term $Q(x, y, t)$ is: $\delta Q(x, y, t) = Q(a) - Q(a_r) = \mathcal{O}(\Delta a_r)$. Since the heat diffusion equation is linear in $Q$, this error translates directly to a temperature deviation of the same order: $\delta T = \mathcal{O}(\Delta a_r)$.

**Temporal integration error:** The PDE simulator advances the solution over a time horizon $\Delta t_r$ using $N_t = \Delta t_r / \Delta t_w$ steps of the forward Euler scheme. Each substep incurs a local truncation error: $\tau_t^{(n)} = \mathcal{O}(\Delta t_w)$, yielding a total accumulated temporal error: $\sum_{n=1}^{N_t} \tau_t^{(n)} = \mathcal{O}(\Delta t_r)$.

Combining the above components, the total one-step prediction error induced by RL interaction with the PDE simulator in the heat diffusion system is given by:

$$\Delta_{\text{RL}} = \underbrace{C_1 \|\Delta\mathbf{x}_{r,\text{int}}\|}_{\text{State observation}} + \underbrace{C_2 \Delta a_r}_{\text{Action discretization}} + \underbrace{C_3 \Delta t_r}_{\text{Temporal integration}} .$$

This decomposition forms the basis for analyzing the interaction fidelity and prediction uncertainty of RL agents in physical systems governed by parabolic PDEs like heat diffusion.

**Refined Total Error Decomposition with Time-Scale Separation.** Based on our prior analysis of surrogate errors in both PDE and RL components, we now present a unified decomposition of all numerical and discretization errors. Over a single RL step, the total prediction error can be expressed as:

$$\Delta_{\text{total}} = \underbrace{C_1 \|\Delta\mathbf{x}_r\|}_{\text{RL (state space)}} + \underbrace{C_2 \Delta a_r}_{\text{RL (action discretization)}} + \underbrace{C_3 \Delta t_r}_{\text{RL (temporal propagation)}} + \underbrace{C_4 \frac{\Delta t_r}{\Delta t_w} (\|\Delta\mathbf{x}_p\| + \Delta a_w)}_{\text{PDE surrogate error, scaled by time resolution}} . \tag{C.11}$$

To ensure the error stays below the acceptable uncertainty level $\Delta y$, we require: $\Delta_{\text{total}} = \mathcal{O}(\Delta y)$. This yields the following matching constraints for all discretization parameters:

$$\|\Delta\mathbf{x}_r\| \sim \|\Delta\mathbf{x}_p\| \sim \Delta a_r \sim \Delta a_w \sim \Delta t_r \sim \Delta y, \quad \text{with} \quad \Delta t_w \ll \Delta y. \tag{C.12}$$

$\rho$-$K$ **Analysis.** We now analyze the prediction error rate in the heat diffusion AI4S environment. We define a prediction error event as one in which the AI4S simulator's output state does not fall within the correct discretized cell of the RL state space. This gives rise to a relative prediction error rate: $\rho = 1 - \frac{\Delta y}{\text{Total Prediction Error}}$, where $\Delta y$ is the size of the RL spatial grid. In a two-dimensional spatial domain (as in heat diffusion over a cooking surface), $\Delta y$ corresponds to the square of the grid length. The total prediction error contains the following components: observation noise, numerical surrogate error and intrinsic error propagation. Over an RL planning horizon $H_r$, the error grows diffusively at rate $\lambda_1$, yielding an error amplification: $\lambda_1 \Delta y / H_r$.

Substituting all components into the definition of $\rho$, we obtain:

$$\rho = 1 - \frac{\Delta y}{\lambda_1 \Delta y / H_r + \Delta y + C_1 \Delta y / K^2} = 1 - \frac{1}{\lambda_1 / H_r + 1 + C_1 / K^2}.$$

So in the limit of high-resolution settings, i.e., when $H_r$ and $K$ are large, we obtain: $\rho = \mathcal{O}\left(\frac{1}{H_r} + \frac{1}{K^2}\right)$. The subsequent results on optimal computational cost allocation are summarized in Table 1 of the main text. As this involves only straightforward calculations and scaling arguments, we omit detailed elaboration here.

### C.5 CART-POLE STABILIZATION

#### C.5.1 ENVIRONMENT DESCRIPTION

In this task, we consider the classic inverted pendulum control problem, where the objective is to apply a horizontal force to a cart in order to maintain a pendulum in an upright and stable position.

The state of the system is represented by a 4-dimensional vector: $\mathbf{x} = \begin{bmatrix} x & \dot{x} & \theta & \dot{\theta} \end{bmatrix}^{\top}$, where $x$ denotes the position of the cart, $\dot{x}$ the velocity of the cart, $\theta$ the angle of the pendulum from the vertical (positive clockwise), and $\dot{\theta}$ the angular velocity of the pendulum. The schematic of the discretization of position and angle in the inverted pendulum environment is shown in Figure C.3.c .

The control input is a horizontal force $u(t) \in [-F_{\max}, F_{\max}]$, which we discretize for reinforcement learning purposes. The agent selects actions from this discretized control space to influence the system dynamics. The physical parameters governing the system include: $M$: the mass of the cart, $m$: the mass of the pendulum, $l$: the length from the pivot to the center of mass of the pendulum, $g$: gravitational acceleration.

Using the Lagrangian formalism, the nonlinear dynamics of the system can be derived as (Nagendra et al., 2017):

$$\ddot{\theta} = \frac{g \sin \theta - \cos \theta \left( \frac{u + ml\dot{\theta}^2 \sin \theta}{M+m} \right)}{l \left( \frac{4}{3} - \frac{m \cos^2 \theta}{M+m} \right)}$$

$$\ddot{x} = \frac{u + ml \left( \dot{\theta}^2 \sin \theta - \ddot{\theta} \cos \theta \right)}{M + m}$$

These equations define the state transition model and control response of the inverted pendulum system. The reinforcement learning agent interacts with this environment by issuing discrete actions at fixed time intervals, aiming to stabilize the pendulum around the upright position. The reward function is typically defined to penalize deviations from the vertical orientation.

#### C.5.2 ERROR COUPLING BETWEEN RL AND ODE SPACES IN AI4S SYSTEMS

Unlike the previous three environments, the control dynamics in the inverted pendulum environment are governed by an ordinary differential equation (ODE) system. But the error analysis procedure remains analogous to that used in the earlier cases. In the following, we analyze the composition of prediction error for the inverted pendulum environment.

**Fine-Grained Surrogate Error in Inverted Pendulum Environments.** We now analyze the surrogate error incurred at each integration step in the inverted pendulum system. Since the dynamics are governed by ordinary differential equations (ODEs), the error sources are reduced to: temporal discretization, state observation through interpolation, and action quantization. We focus on the position variable $x(t)$ of the cart and quantify its one-step prediction error.

**Temporal truncation error:** The ODE describing the cart's horizontal motion is: $\ddot{x} = f(\mathbf{x}, u)$, where $u \in [-F_{\max}, F_{\max}]$ is the external force and $\mathbf{x} = [x, \dot{x}, \theta, \dot{\theta}]^{\top}$ is the state vector.

Using forward Euler integration for both velocity and position: $\dot{x}^{n+1} = \dot{x}^n + \Delta t_w \ddot{x}^n, \quad x^{n+1} = x^n + \Delta t_w \dot{x}^n$, the local truncation error from Taylor expansion becomes: $x^{n+1} = x^n + \Delta t_w \dot{x}^n + \frac{\Delta t_w^2}{2} \ddot{x}^n + \mathcal{O}(\Delta t_w^3)$. Then the numerical derivative satisfies: $\frac{x^{n+1} - x^n}{\Delta t_w} = \dot{x}^n + \frac{1}{2} \Delta t_w \ddot{x}^n + \mathcal{O}(\Delta t_w^2)$.

From the expression of $\ddot{x}$ in the inverted pendulum dynamics, we know that $\ddot{x}^n$ is bounded. Hence, we obtain the temporal truncation error: $\tau_{\text{time}} = \mathcal{O}(\Delta t_w)$.

**State discretization error :** The state is often represented in a coarsely discretized state space. Suppose the cart's true position $x \in \mathbb{R}$ is approximated by a discrete cell center $\tilde{x}$. Using linear interpolation or nearest-neighbor projection, the induced error is: $|x - \tilde{x}| = \mathcal{O}(\|\Delta \mathbf{x}_p\|)$, which propagates into the dynamics via the nonlinear function $f(\mathbf{x}, u)$. Assuming Lipschitz continuity of $f$, this implies: $\Delta_x^{\text{state}} = \mathcal{O}(\|\Delta \mathbf{x}_p\|)$.

**Action-induced control error:** The control force $u$ is discretized into a finite set, and the applied value $u_w$ approximates the true value with resolution $\Delta a$. The dynamics respond linearly to changes in $u$, as seen from the equation for $\ddot{x}$, so: $\Delta_x^{\text{action}} = \mathcal{O}(\Delta a_w)$.

**Final per-step surrogate error:** Summing all components, the total one-step surrogate error in predicting the cart position is: $\Delta_x^{(k)} = \mathcal{O}(\Delta t_w) + \mathcal{O}(\|\Delta \mathbf{x}_p\|) + \mathcal{O}(\Delta a_w)$.

**One-Step Prediction Error in RL-Based Inverted Pendulum Control.** In this task, we analyze the surrogate prediction error that arises when an RL agent interacts with the dynamical system of an inverted pendulum. The dynamics are governed by a nonlinear second-order ordinary differential equation (ODE), and we focus our analysis on the position variable $x(t)$, which is affected by both control input and physical evolution. The prediction error consists of three major sources: temporal discretization, spatial resolution (observation) error, and control discretization error.

**Temporal integration error:** The ODE simulator advances the solution over a time horizon $\Delta t_r$ using $N_t = \Delta t_r / \Delta t_w$ steps of the forward Euler scheme. Each substep incurs a local truncation error: $\tau_t^{(n)} = \mathcal{O}(\Delta t_w)$, yielding a total accumulated temporal error: $\sum_{n=1}^{N_t} \tau_t^{(n)} = \mathcal{O}(\Delta t_r)$.

**Observation-induced spatial error:** Suppose the RL agent does not access the true continuous state $x(t_n)$, but instead receives a quantized observation $\tilde{x}(t_n)$, derived from uniform discretization over spatial grid resolution $\Delta x_r$. In one-dimensional observation with linear interpolation or rounding, the state projection satisfies: $|x(t_n) - \tilde{x}(t_n)| \leq \frac{1}{2}\Delta x_r$. This discrepancy is formally equivalent to a spatial interpolation error and yields a spatial state uncertainty: $\tau_{\text{state}} = \mathcal{O}(\Delta x_r)$.

**Action discretization error:** The control input $u$ is issued from a discretized action set $a_r \in \mathcal{A}_r$, such that: $u(t_n) = \bar{u}(t_n) + \delta u(t_n)$, $\delta u = \mathcal{O}(\Delta a_r)$, where $\bar{u}(t_n)$ is the true continuous control signal and $\delta u$ is the discretization mismatch. From the expression of $\ddot{x}$, this control deviation contributes linearly to the evolution of position through: $\delta x^{n+1} = \Delta t_r \delta \dot{x}^{n+1} = \Delta t_r^2 \cdot \frac{\delta u}{M+m} = \mathcal{O}(\Delta a_r)$. We treat time as a fixed constant when integrating the accumulated effect of the action, rather than considering it as an error term.

**Total error:** Combining the above contributions from temporal, spatial, and control discretization errors, the one-step prediction error in the inverted pendulum environment satisfies:

$$\Delta_{\text{pendulum}} = \underbrace{C_1 \Delta x_r}_{\text{State resolution}} + \underbrace{C_2 \Delta a_r}_{\text{Action resolution}} + \underbrace{C_3 \Delta t_r}_{\text{Temporal integration}} .$$

**Refined Total Error Decomposition with Time-Scale Separation.** Based on our previous analysis of surrogate error in both the ODE and RL components, we now integrate all sources of discretization and numerical error into a unified framework. Over a single RL step, the total prediction error can be decomposed as follows:

$$\Delta_{\text{total}} = \underbrace{C_1 \|\Delta \mathbf{x}_r\|}_{\text{RL (state space)}} + \underbrace{C_2 \Delta a_r}_{\text{RL (action discretization)}} + \underbrace{C_3 \Delta t_r}_{\text{RL (temporal propagation)}} + \underbrace{C_4 \frac{\Delta t_r}{\Delta t_w}(\|\Delta \mathbf{x}_p\| + \Delta a_w)}_{\text{PDE surrogate error, scaled by time resolution}} .$$

$$(\text{C.13})$$

To ensure that the learned RL transition kernel remains robust under observation uncertainty $\Delta y$, we require that: $\Delta_{\text{total}} = \mathcal{O}(\Delta y)$. This constraint implies a resolution matching condition across all discretization dimensions. Specifically, we obtain the following asymptotic scaling relations:

$$\|\Delta \mathbf{x}_r\| \sim \|\Delta \mathbf{x}_p\| \sim \Delta a_r \sim \Delta a_w \sim \Delta t_r \sim \Delta y, \quad \text{with} \quad \Delta t_w \ll \Delta y. \qquad (\text{C.14})$$

**$\rho$-$K$ Analysis.** We now analyze the prediction error rate in the cart-pole environment. The total prediction error contains the following components: observation noise, numerical surrogate error and intrinsic error propagation. Over an RL planning horizon $H_r$, the error grows diffusively at rate $\lambda_1$, yielding an error amplification: $\lambda_1 \Delta y / H_r$. Substituting all components into the definition of $\rho$, we obtain:

$$\rho = 1 - \frac{\Delta y}{\lambda_1 \Delta y / H_r + \Delta y + C_1 \Delta y / K} = 1 - \frac{1}{\lambda_1 / H_r + 1 + C_1 / K}.$$

Since there is no notion of state grids in the inverted pendulum environment and all state discretizations are first-order, the order of $K$ is one. So in the limit of high-resolution settings, i.e., when $H_r$ and $K$ are large, we obtain: $\rho = \mathcal{O}\left(\frac{1}{H_r} + \frac{1}{K}\right)$. The subsequent results on optimal computational cost allocation are summarized in Table 1 of the main text. As this involves only straightforward calculations and scaling arguments, we omit detailed elaboration here.

### C.5.3 EMPIRICAL COMPARISON OF TABULAR RL AND DEEP Q-LEARNING

We designed an experiment to test our theoretical findings on different learning algorithms. Using a data-driven inverted pendulum (CartPole), we compared the sample complexity of Tabular RL and Q-Learning. This comparison was performed under various discrete observation resolutions.

**Experimental Setup** The CartPole system consists of a cart moving along a horizontal track with a hinged pole on top. The agent's goal is to apply horizontal forces to maintain the pole in an upright position. We adopt a two-stage workflow: (i) state-action-next-state tuples are collected from a physics-based simulator and used to train an MLP-based dynamics model, and (ii) this learned model serves as the environment for RL training.

**Reward Function** To provide smooth learning signals and mitigate oscillations, we employ a shaped reward function:

$$r_t = \begin{cases} 1 - \alpha|\theta_t| - \beta|\dot{\theta}_t| & \text{if } |\theta_t| \leq 12° \text{ and } |x_t| \leq 2.4 \\ 0 & \text{otherwise} \end{cases} \tag{C.15}$$

where $\theta_t$ and $\dot{\theta}_t$ are the pole angle and angular velocity, respectively. The coefficients $\alpha, \beta > 0$ penalize deviation from the vertical and excessive oscillation. An episode terminates with zero reward if the pole falls or the cart leaves the track.

**Learning Methods** We train two agents under identical discretized observation resolutions:

- **Tabular RL:** Implemented using the value iteration algorithm, with environment dynamics derived from physics-based simulation and interpolated to the RL observation space, without neural network approximation.

- **Q-Learning:** Implemented using a Deep Q-Network (DQN) with discretized state and action spaces.

Performance is measured by the number of samples required to achieve a value function error below 1% and the corresponding wall-clock runtime on a single NVIDIA RTX 4090 GPU.

**Results** The detailed experimental results are presented in Table C.3 and Table C.4. The results show that Q-Learning is significantly more sample-efficient than tabular RL, which is expected due to the generalization capability of the neural network. However, both methods show a clear sensitivity to the resolution parameters $H_r$ and $K$. Optimal performance is not achieved at the highest or lowest resolutions but rather at an intermediate point, which is consistent with our theoretical analysis. For tabular RL, the optimal configuration is found at $K = 1.5$ and $\log_{H_w} H_r = 1/3$. For Q-Learning, the optimum shifts to $K = 2.0$ and $\log_{H_w} H_r = 1/2$, suggesting that DQN can better leverage a higher-fidelity physical model. These results strongly support our central claim that even for deep RL methods, the resolutions of the physical simulation and the RL algorithm must be carefully coordinated to achieve optimal performance.

## D FUNCTIONAL ANALYSIS TOOLS FOR PDE ERROR AND LIMIT ANALYSIS

In this section, we provide a collection of classical results from functional analysis and PDE theory that underpin the error analysis, convergence behavior, and design of discretization schemes in AI4S-RL environments governed by PDEs (Evans, 2022; E, 2011). These tools help quantify how physical solution properties interact with discretized approximations, especially under limited regularity, complex boundaries, and dynamic state evolution.

Table C.3: Cart-Pole System: Sample Complexity $\log_{10}(N)$ for Tabular RL to Achieve Value Function Error $< 0.01$.

| $\log_{H_w} H_r$ | K=1.0 | K=1.5 | K=2.0 | K=2.5 |
|---|---|---|---|---|
| 1/8 | 4.51(4.58/4.45) | 4.50(4.53/4.46) | 4.51(4.57/4.44) | 4.54(4.61/4.48) |
| 1/6 | 4.26(4.32/4.19) | 4.25(4.31/4.18) | 4.26(4.30/4.23) | 4.29(4.36/4.23) |
| 1/5 | 4.10(4.17/4.04) | 4.09(4.13/4.06) | 4.10(4.16/4.03) | 4.14(4.21/4.08) |
| 1/4 | 3.91(3.98/3.85) | 3.90(3.94/3.87) | 3.91(3.95/3.88) | 3.94(4.01/3.88) |
| 1/3 | 3.77(3.81/3.74) | **3.65(3.71/3.58)** | 3.74(3.80/3.67) | 3.88(3.95/3.82) |
| 1/2 | 3.95(4.02/3.89) | 3.83(3.87/3.80) | 3.92(3.99/3.86) | 4.06(4.13/3.99) |
| 1 | 4.25(4.29/4.22) | 4.13(4.20/4.07) | 4.22(4.26/4.19) | 4.36(4.42/4.29) |
| 2 | 4.55(4.62/4.49) | 4.43(4.47/4.40) | 4.52(4.59/4.46) | 4.66(4.72/4.59) |

Table C.4: Cart-Pole System: Sample Complexity $\log_{10}(N)$ for Q-Learning to Achieve Value Function Error $< 0.01$.

| $\log_{H_w} H_r$ | K=1.0 | K=1.5 | K=2.0 | K=2.5 |
|---|---|---|---|---|
| 1/8 | 3.20(3.26/3.13) | 2.95(3.01/2.88) | 2.82(2.86/2.79) | 2.90(2.97/2.84) |
| 1/6 | 3.05(3.11/2.98) | 2.80(2.84/2.77) | 2.72(2.78/2.65) | 2.80(2.86/2.73) |
| 1/5 | 2.90(2.94/2.87) | 2.75(2.81/2.68) | 2.68(2.72/2.65) | 2.75(2.82/2.69) |
| 1/4 | 2.80(2.87/2.74) | 2.70(2.74/2.67) | 2.67(2.73/2.61) | 2.72(2.76/2.69) |
| 1/3 | 2.78(2.85/2.72) | 2.69(2.73/2.66) | 2.66(2.72/2.59) | 2.71(2.78/2.65) |
| 1/2 | 2.75(2.79/2.72) | 2.67(2.73/2.60) | **2.65(2.69/2.62)** | 2.67(2.74/2.61) |
| 1 | 2.85(2.91/2.78) | 2.78(2.82/2.75) | 2.78(2.85/2.72) | 2.82(2.86/2.79) |
| 2 | 3.10(3.17/3.04) | 2.95(2.99/2.92) | 2.93(2.98/2.87) | 3.00(3.07/2.94) |

### D.1 MOTIVATION: WHY DISCRETE FORMATS AND FUNCTION SPACES MATTER IN AI4S-RL

In PDE-governed AI4S-RL tasks, control and prediction rely on the accurate approximation of continuous state evolution. However, neural networks or other function approximators inherently operate over discrete representations. Therefore, the error between the true PDE solution and the learned approximation depends not only on the architecture but also on how the discretization aligns with the analytical properties of the solution. Tools from functional analysis allow us to assess:

- How regular or irregular the true solution is (e.g., near boundaries or under shocks);
- How the discretization grid or function basis must adapt to preserve convergence;
- How stability and generalization are affected by boundary singularities and approximation limitations.

The following theorems provide essential insights into these questions and can be naturally organized along a logical chain used in theoretical PDE analysis:

- Start with basic control over functions via gradients (Poincaré Inequality),
- Link smoothness to representability and generalization (Sobolev Embedding),
- Ensure PDE well-posedness (Lax-Milgram),
- Connect numerical approximations to best approximants (Céa's Lemma),
- Use compactness to show convergence under refinement (Rellich-Kondrachov),
- Extend to temporal-spatial regularity (Aubin-Lions),
- Analyze boundary effects (Maximum Principle, Trace Theorem),
- Control physical evolution through energy bounds (Energy Estimates).

The materials presented in this section are well-established results in computational mathematics and are not original contributions by the authors. Our aim is to provide readers—particularly those interested in the error analysis of PDE-based surrogates—with an introductory overview of the

foundational knowledge in this field. We emphasize concepts essential for understanding boundary error behavior in PDEs, especially those involving the trace theorem, which is repeatedly applied throughout this paper.

## D.2 CORE THEOREMS AND THEIR ROLES

**1. Poincaré Inequality.** Let $u \in H_0^1(\Omega)$ on a bounded domain $\Omega \subset \mathbb{R}^d$. Then:

$$\|u\|_{L^2(\Omega)} \leq C\|\nabla u\|_{L^2(\Omega)}.$$

**Interpretation:** This inequality ensures that if the gradient of a function is small, then the function itself must also be small in the $L^2$ sense. It captures how global behavior can be controlled through local derivatives. **Use:** In AI4S-RL systems, this inequality provides the foundation for energy-based bounds, helping ensure that learned control policies do not cause unbounded or physically unrealistic growth in system states during training or rollout (Evans, 2022).

**2. Sobolev Embedding Theorem.** For sufficiently smooth domains and appropriate $s$, $H^s(\Omega) \hookrightarrow L^p(\Omega)$ or $C^k(\Omega)$.
**Interpretation:** This theorem allows one to infer integrability or continuity from Sobolev space membership, linking smoothness to spatial regularity. **Use:** In AI4S-RL, this theorem helps justify the use of neural approximators, showing under what conditions a function learned in a weak (Sobolev) sense can be expected to generalize as a strong or continuous function (Evans, 2022).

**3. Lax-Milgram Theorem.** Given a bounded, coercive bilinear form $a(\cdot, \cdot)$ on a Hilbert space $V$, the variational problem:

$$a(u, v) = f(v), \quad \forall v \in V,$$

has a unique solution $u \in V$.
**Interpretation:** This theorem guarantees that weak formulations of PDEs are well-posed under mild assumptions. **Use:** It is critical for framing AI4S-RL problems involving constrained learning, such as solving inverse or control problems governed by PDEs. Ensures the existence and uniqueness of target states (Evans, 2022).

**4. Céa's Lemma.** Let $u$ solve the variational problem and $u_h \in V_h \subset V$ be the approximation. Then:

$$\|u - u_h\|_V \leq \frac{C}{\alpha} \inf_{v_h \in V_h} \|u - v_h\|_V.$$

**Interpretation:** This lemma asserts that the best possible error of a numerical approximation is bounded proportionally by the best projection error. **Use:** In AI4S-RL, this guides the choice of representation spaces (e.g., basis functions, neural network architectures) and ensures that approximate policies or value functions can converge quasi-optimally (Hou, 2003).

**5. Rellich–Kondrachov Compactness Theorem.** If $\Omega$ is bounded and $u_n$ is bounded in $H^1(\Omega)$, then a subsequence converges strongly in $L^2(\Omega)$.
**Interpretation:** This compactness result ensures that bounded sequences of approximate solutions have strongly convergent subsequences. **Use:** It provides a theoretical foundation for showing convergence of learned policies or solutions in AI4S-RL as the discretization or model resolution is refined (Evans, 2022).

**6. Aubin–Lions Lemma.** Let $X_0 \hookrightarrow X \hookrightarrow X_1$ be Banach spaces with compact embedding. If $u_n$ is bounded in $L^p(0, T; X_0)$ and $\partial_t u_n$ is bounded in $L^q(0, T; X_1)$, then $u_n$ is relatively compact in $L^p(0, T; X)$.
**Interpretation:** This lemma bridges temporal and spatial regularity, ensuring strong convergence when time derivatives and spatial norms are controlled. **Use:** Vital in PDE-driven RL settings with temporal evolution. For instance, it helps demonstrate that learned state sequences from policy iteration converge to true continuous dynamics (Lions, 1969).

**7. Maximum Principle.** Let $u$ solve an elliptic or parabolic PDE with suitable structure. Then:

$$\max_{\Omega} u = \max_{\partial\Omega} u.$$

**Interpretation:** This principle ensures that solution extremums occur on boundaries, preserving physical constraints. **Use:** In AI4S-RL, this is crucial when dealing with bounded domains and enforcing reward shaping or physical limits. Helps constrain exploration near boundary zones (Majda et al., 2002).

**8. Energy Estimates.** Multiply a PDE by $u$ or $\partial_t u$, integrate over space-time, and derive:

$$E(t) = \int_{\Omega} \left( |\partial_t u|^2 + |\nabla u|^2 \right) dx \quad \text{is bounded or decaying.}$$

**Interpretation:** These estimates show that system energy either remains bounded or decreases over time. **Use:** Critical for stability analysis of RL agents in physical systems. Ensures that learned dynamics don't lead to energy blow-up or unrealistic oscillations (E, 2011; Majda et al., 2002).

**9. Trace Theorem.** For $u \in H^1(\Omega)$, its restriction to the boundary $\partial\Omega$ satisfies:

$$\gamma(u) := u|_{\partial\Omega} \in H^{1/2}(\partial\Omega).$$

**Interpretation:** Functions in Sobolev spaces lose half a derivative when restricted to boundaries. **Use:** This loss of smoothness explains why control near boundaries is harder in AI4S-RL systems and why special handling is needed in boundary-constrained reward or control definitions (Evans, 2022).

### D.3 LINKING THEORY TO PRACTICE

These tools collectively help answer: How should one discretize state, design control inputs, and construct reward functionals when the solution is only partially smooth, or when boundary behavior dominates the error? Trace and Poincaré help identify when boundary refinement is necessary; Sobolev embedding and Rellich-Kondrachov justify the use of compact approximators; Lax-Milgram and Céa give the mathematical foundation for why value function approximators can converge. Understanding and incorporating these results into the AI4S-RL model design allows for principled control over generalization, robustness, and physical consistency in the learning loop.

## E EXPERIMENTAL DETAILS: TEPPANYAKI THERMODYNAMIC CONTROL MODEL

### E.1 PROBLEM FORMULATION AND PHYSICAL MODEL

#### E.1.1 PHYSICAL SYSTEM DESCRIPTION

We construct a teppanyaki temperature control simulation based on the two-dimensional unsteady heat conduction partial differential equation. The computational domain is a uniform square metal plate $\Omega = [0,1] \times [0,1]$ m$^2$. The system is equipped with $H_s = 2$ power-adjustable heat sources. These sources are fixed at spatial positions $\mathbf{x}_s^{(1)} = (0.25, 0.5)$ and $\mathbf{x}_s^{(2)} = (0.75, 0.5)$. The control variables are the heating powers $P_j(t) \in [0, 1000]$ W for each source.

Three dishes are placed on the plate. Each dish $i$ is modeled as a square region $\mathcal{D}_i \subset \Omega$ with side length $L_d = 0.1$ m. Their geometric centers are located at $\mathbf{x}_d^{(1)} = (0.4, 0.5)$, $\mathbf{x}_d^{(2)} = (0.5, 0.5)$, and $\mathbf{x}_d^{(3)} = (0.6, 0.5)$. The target temperatures are $\{70, 80, 90\}$°C, and the cooking durations are $\{50, 55, 60\}$ seconds. The control objective is to minimize the average temperature deviation of all dishes within their cooking windows over a total duration of $T_{total} = 60$ seconds.

The regional average temperature of dish $i$ at time $t$ is defined as:

$$\bar{T}_i(t) = \frac{1}{|\mathcal{D}_i|} \iint_{\mathcal{D}_i} T(\mathbf{x}, t) \, d\mathbf{x} \tag{E.1}$$

The global loss function is defined as the cumulative mean squared error:

$$J = \int_0^{T_{total}} \sum_{i=1}^{N_d} \mathbb{I}_{[0,t_i^{cook}]}(t) \cdot \left(\bar{T}_i(t) - T_i^{target}\right)^2 dt \tag{E.2}$$

Here $\mathbb{I}_{[0,t_i^{cook}]}(t)$ is a time indicator function. This function ensures that temperature errors are computed only during the dish placement period.

### E.1.2 GOVERNING EQUATIONS

The spatiotemporal evolution of the temperature field $T(\mathbf{x}, t)$ follows the two-dimensional inhomogeneous heat conduction equation:

$$\frac{\partial T}{\partial t} = \alpha \nabla^2 T + Q(\mathbf{x}, t), \quad \mathbf{x} \in \mathbb{R}^2, \, t > 0 \tag{E.3}$$

where $\alpha = 5 \times 10^{-4}$ m$^2$/s is the thermal diffusivity coefficient. The boundary condition adopts an infinite domain assumption: $\lim_{|\mathbf{x}| \to \infty} T(\mathbf{x}, t) = T_\infty$. The initial condition is set to the ambient temperature $T(\mathbf{x}, 0) = T_\infty = 20°C$. The heat source term $Q(\mathbf{x}, t)$ is constructed by superposing $H_s$ Gaussian-distributed heating elements:

$$Q(\mathbf{x}, t) = \sum_{j=1}^{H_s} \frac{P_j(t)}{\rho c_p \cdot 2\pi\sigma^2} \exp\left(-\frac{|\mathbf{x} - \mathbf{x}_s^{(j)}|^2}{2\sigma^2}\right) \tag{E.4}$$

Here $\sigma = 0.05$ m is the effective radius of each heat source. The material properties are density $\rho = 7850$ kg/m$^3$ and specific heat capacity $c_p = 460$ J/(kg·K).

## E.2 MULTI-RESOLUTION REINFORCEMENT LEARNING ENVIRONMENT

### E.2.1 SPATIOTEMPORAL DISCRETIZATION SCHEME

We employ a high-precision uniform Cartesian grid to partition the physical domain $\Omega$ for bottom-layer physical evolution computation. The physical grid spacing is set to $\Delta x_{phy} = 0.025$ m. This corresponds to $N_{phy} \times N_{phy} = 40 \times 40$ grid cells. The physical time step is $\Delta t_{phy} = 1/450$ s $\approx 0.0022$ s. This satisfies the CFL stability condition for explicit finite difference schemes.

The coarsening degree of the RL observation space is controlled by parameter $K \geq 1$. The RL grid size is $N_{rl} = \lfloor N_{phy}/K \rfloor$ with grid spacing $\Delta x_{rl} = K \cdot \Delta x_{phy}$. Spatial coarsening is implemented through regional averaging:

$$T_{rl}^{(i,j)} = \frac{1}{K^2} \sum_{p=0}^{K-1} \sum_{q=0}^{K-1} T_{phy}^{(Ki+p,Kj+q)} \tag{E.5}$$

where $T_{phy}^{(i,j)}$ represents the temperature values on the physical grid. These values are obtained by evaluating the analytical solution at the corresponding spatial coordinates $(i\Delta x_{phy}, j\Delta x_{phy})$.

The RL decision time step $\Delta t_{rl}$ is determined by parameter $y = \log_{h_w}(h_r)$, where $h_w = T_{total}/\Delta t_{phy}$ is the temporal horizon at the physical layer and $h_r = h_w^y$ is the horizon at the RL layer. This design constructs a hierarchical simulation structure. The underlying physical environment evolves $H = \lfloor \Delta t_{rl}/\Delta t_{phy} \rfloor$ micro-steps for each decision executed by the agent. Different combinations of parameters $K$ and $y$ define 25 experimental configurations. The trade-off between sample efficiency and computational complexity across these configurations is the core focus of this study.

### E.2.2 STATE-ACTION-REWARD MECHANISM

The system state $\mathbf{s}$ consists of the global temperature field $T_{rl} \in \mathbb{R}^{N_{rl} \times N_{rl}}$ on the coarsened grid at the current time, along with the normalized remaining cooking time $\tilde{t}_i$ for each food item. Parameter $K$ defines the coarse-graining degree of observations. Temperature readings are discretized at intervals of $\Delta T_{disc} = 2°C \times K$. As $K$ increases, the agent faces dual challenges of spatial information

blurring (pixelation) and temperature quantization noise. These factors significantly intensify partial observability of the environment. The action space degrades correspondingly with the state space. Power adjustment of the two heat sources is modeled as discrete actions. The adjustment granularity is directly coupled to $K$ through the minimum power increment $\Delta P = 50\,\text{W} \times K$. This coupling causes the number of selectable power levels to decrease significantly as $K$ increases. Coarse grids correspond to coarse control, which aligns with physical intuition.

To ensure consistency and physical meaning of the reward function across different time steps $\Delta t_{rl}$, we adopt a continuous integration form based on physical micro-steps. Within the $k$-th RL decision step, the instantaneous reward $r_k$ is computed as the cumulative temperature error over all physical micro-steps in that period:

$$r_k = \sum_{h=1}^{H} \left[ - \sum_{i=1}^{Num_d} \mathbb{I}_{[0, t_i^{cook}]}(t_{k,h}) \cdot \frac{(\bar{T}_i(t_{k,h}) - T_i^{target})^2}{T_{scale}^2} \right] \cdot \Delta t_{phy} \tag{E.6}$$

where $H = \lfloor \Delta t_{rl}/\Delta t_{phy} \rfloor$ is the number of physical micro-steps within a single RL step, and $T_{scale} = 50°\text{C}$ is the normalization constant. Through this approach, the cumulative reward $R = \sum r_k$ remains an unbiased approximation of the original continuous control objective equation E.2, regardless of variations in RL decision frequency. This design guarantees fairness in computational cost comparisons across different experimental groups.

### E.3    TRAINING ALGORITHM AND CONVERGENCE CRITERIA

#### E.3.1    PPO ALGORITHM IMPLEMENTATION

We adopt Proximal Policy Optimization (PPO) as the core training algorithm, considering the mixed characteristics of continuous state space and discrete action space in this problem. Compared to off-policy algorithms, PPO constrains policy update step sizes through clipped objective functions. This mechanism effectively prevents policy oscillations in complex non-stationary thermodynamic environments and ensures training robustness. Both the policy network (Actor) and value network (Critic) employ multi-layer perceptron (MLP) architectures with the global state vector **s** as input. The hyperparameters are set as follows: learning rate $3 \times 10^{-4}$, discount factor $\gamma = 0.99$, and Generalized Advantage Estimation (GAE) parameter $\lambda = 0.95$. Training proceeds synchronously across 8 parallel environments with GPU-accelerated gradient updates.

#### E.3.2    CONVERGENCE CRITERIA

To fairly compare sample efficiency across different resolution configurations $(K, y)$, we adopt a relative convergence criterion. Training is considered converged and terminated when the agent achieves the preset performance threshold in three consecutive evaluations, where each evaluation is based on the average over 10 complete episodes. The specific threshold is set to a theoretically derived optimal control baseline (see Section 3 for the physical environment setup). This threshold setting ensures that computational cost comparisons across configurations are meaningful under comparable performance levels.

### E.4    EXPERIMENTAL CONFIGURATION AND RESULTS

The experiment covers spatial resolution ratios $K \in \{1.0, 2.0, 4.0, 6.0, 8.0\}$ and temporal resolution parameter $y = \log_{h_w}(h_r) \in \{1/8, 1/6, 1/4, 1/3, 1/2\}$, totaling 25 configuration points. Each configuration is trained with five random seeds until reaching the convergence criterion. All experiments are executed on the same hardware platform (Intel Xeon Gold 6530, NVIDIA RTX 4090, 256 GB RAM).

#### E.4.1    $\varepsilon$–$N$ SCALING ANALYSIS

To validate the theoretical $\varepsilon$–$N$ scaling predictions, we analyze complete scaling curves for all 25 resolution configurations. Here, $\varepsilon$ denotes the suboptimality gap defined as $\varepsilon = R^* - R_{\text{current}}$, where $R^*$ is the optimal policy reward derived from the analytical solution. The variable $N$ represents training steps.

Panel (c) of Figure 6 shows $\varepsilon$–$N$ scaling curves for all 23 configurations that successfully reach the convergence threshold. Different colors represent temporal resolution $y$ values, while color intensity indicates spatial resolution $K$ (darker shades for smaller $K$, lighter for larger $K$). Shaded regions illustrate the performance range across different $K$ values within each temporal resolution group. Vertical dashed lines mark the mean convergence step for each $y$, revealing clear separation between temporal resolution groups: finer temporal resolution (smaller $y$) requires fewer training steps to converge.

### E.4.2 COMPUTATIONAL COST DECOMPOSITION

A key contribution of our experimental analysis is the rigorous decomposition of total computational cost into its constituent components. In AI4S reinforcement learning systems, computational resources are consumed by two fundamentally different processes: (1) physical simulation on CPU, which involves solving or evaluating PDE-based environment dynamics, and (2) neural network operations on GPU, which includes policy inference and gradient-based training updates.

This decomposition is critical because CPU and GPU computational costs scale differently with resolution parameters. CPU costs are dominated by the complexity of the physical solver, which depends on grid resolution and time step constraints imposed by numerical stability. GPU costs are primarily determined by the number of RL training iterations required for convergence, which depends on the effective state-action space size and the difficulty of credit assignment under different temporal horizons.

Panels (d), (e), and (f) of Figure 6 present heatmaps showing the CPU, GPU, and total computational costs respectively across all 25 configurations. Several important observations emerge from this decomposition:

**CPU-GPU trade-off structure.** The CPU and GPU cost landscapes exhibit distinct patterns. CPU costs tend to increase with finer temporal resolution (smaller $\log_{h_w}(h_r)$) due to the larger number of physical simulation steps required. GPU costs show more complex dependence on both spatial and temporal parameters, reflecting the interplay between state space complexity and horizon length in the RL training dynamics.

**Non-additive composition.** The total cost is not simply the sum of CPU and GPU costs in logarithmic scale. The relative contribution of each component varies significantly across the configuration space. At fine resolutions, CPU costs dominate due to expensive physical simulations. At coarse resolutions, GPU costs become relatively more significant as the RL algorithm struggles with reduced observability and must compensate through additional training iterations.

**Optimal configuration identification.** The optimal configuration ($K = 6.0$, $\log_{h_w}(h_r) = 1/3$) achieves the minimum total cost by balancing these competing effects. This configuration provides sufficient resolution for the agent to learn effective control policies while avoiding the computational overhead of unnecessarily fine discretization.

### E.4.3 COMPUTATIONAL COST SUMMARY

For each configuration, the logical interaction count $N_{\text{total}}$ recorded by environment destructors is combined with the sampling multiplier $M_{\text{sample}}(K, \Delta t)$ measured from post-training sampling experiments (see Section E.5). The equivalent total computational cost is reconstructed in units of $\log_{10}(\text{FLOPs}_{\text{total}})$. Detailed numerical data with confidence intervals are presented in Table E.5.

Table E.5: Equivalent Total Computational Cost ($\log_{10}(\text{FLOPs}_{\text{total}})$) with Confidence Intervals

| $\log_{h_w}(h_r)$ | $K = 1$ | $K = 2$ | $K = 4$ | $K = 6$ | $K = 8$ |
|---|---|---|---|---|---|
| 1/8 | $17.77 \pm 0.29$ | $17.74 \pm 0.35$ | $17.72 \pm 0.26$ | $17.77 \pm 0.14$ | N/A |
| 1/6 | $17.59 \pm 0.42$ | $17.60 \pm 0.29$ | $17.65 \pm 0.21$ | $17.65 \pm 0.41$ | N/A |
| 1/4 | $17.59 \pm 0.26$ | $17.56 \pm 0.13$ | $17.50 \pm 0.17$ | $17.54 \pm 0.19$ | $17.81 \pm 0.27$ |
| 1/3 | $17.64 \pm 0.17$ | $17.59 \pm 0.22$ | $17.41 \pm 0.20$ | $\mathbf{17.27 \pm 0.22}$ | $17.69 \pm 0.10$ |
| 1/2 | $17.60 \pm 0.15$ | $17.70 \pm 0.19$ | $17.58 \pm 0.20$ | $17.36 \pm 0.08$ | $18.00 \pm 0.08$ |

**Note:** Values represent $\log_{10}(\text{FLOPs}_{\text{total}})$ with standard deviation computed from five random seeds. Both the logical interaction count $N_{\text{total}}$ during training and the statistical sampling multiplier $M_{\text{sample}}$ are comprehensively considered. Configurations marked **N/A** fail to reach the performance threshold due to excessive discretization coarseness. Specifically, $K = 8.0$ corresponds to only $5 \times 5$ RL observation grids and action resolution of $\Delta P = 400$ W (with merely 3 selectable levels: 0 W, 400 W, 800 W). Under this granularity, the agent cannot achieve fine temperature control. The optimal configuration ($K = 6.0, \log_{h_w}(h_r) = 1/3$) is highlighted in bold, corresponding to the lowest equivalent computational complexity of $10^{17.27 \pm 0.22}$ FLOPs. This result confirms the theoretical prediction that both excessively coarse and excessively fine discretization lead to efficiency losses.

To ensure the reliability of our conclusions, we conducted extensive statistical analysis across multiple random seeds. Each configuration was trained with five independent random seeds, controlling for variations in network initialization, environment stochasticity, and optimization trajectory. The confidence intervals reported in Table E.5 and visualized in Figure 6 represent the standard deviation of the total computational cost across these runs.

### E.5 COMPUTATIONAL COST RECONSTRUCTION METHODOLOGY

To ensure that experimental conclusions generalize to real AI4S scenarios where numerical PDE solvers must be used, we adopt a computational cost reconstruction approach. The core challenge is that directly recording the CPU/GPU execution time for every state transition during training is complex and implementation-dependent. Specifically, each invocation of the environment's physics computation (via destructor calls for object lifecycle management) may require vastly different convergence iterations depending on local condition numbers of the linear systems or nonlinear solver states at different grid cells. Recording these fine-grained execution times would require instrumenting every grid cell's solver, which is impractical for large-scale experiments.

Instead, we adopt a two-stage approach: first, we complete the full PPO training for all 25 resolution configurations and record only the *logical interaction count* $N_{\text{total}}$ via destructor counters in the environment core class. This count reflects the sample complexity of the RL algorithm itself, independent of the underlying physics solver implementation. Second, after training converges, we perform a post-hoc sampling study to estimate the *computational multiplier* that would be required in real scenarios using numerical solvers.

Specifically, we randomly select a representative subset of state-action pairs encountered during converged policy execution. For each selected state, we perform Monte Carlo sampling within the observation uncertainty range $[\mathbf{x} - \Delta y/2, \mathbf{x} + \Delta y/2]$ to measure the frequency distribution of the next state falling into different grid cells. According to Theorem 1 in the main text, the number of samples $M_{\text{sample}}(K, \Delta t)$ required to statistically distinguish the correct target grid with confidence $1 - \delta$ depends on the classification margin:

$$M_{\text{sample}}(K, \Delta t) = \mathcal{O}\left(\frac{\log(1/\delta)}{\min_j(\Delta p^{(j)})^2}\right) \tag{E.7}$$

where $\Delta p^{(j)}$ is the probability gap between the dominant grid cell and its closest competitor in dimension $j$.

We then estimate the per-sample computational cost $C_{\text{solver}}(K)$ for a single PDE solver call at resolution $K$. For explicit finite difference methods with grid size $N_{phy}/K$ and CFL-constrained time step, this cost scales as $C_{\text{solver}}(K) \sim (N_{phy}/K)^d$ per time step, where $d$ is the spatial dimension. The final reconstructed total cost is:

$$\text{FLOPs}_{\text{total}}(K, \Delta t) = N_{\text{total}} \times M_{\text{sample}}(K, \Delta t) \times C_{\text{solver}}(K) \tag{E.8}$$

This reconstruction methodology ensures that reported computational costs honestly reflect what would be required in real AI4S applications using numerical solvers, while still enabling efficient large-scale hyperparameter searches during the training phase. The sampling multiplier $M_{\text{sample}}$ accounts for the statistical overhead of state disambiguation under observation noise, and the solver cost $C_{\text{solver}}$ accounts for the grid-resolution-dependent computational burden. Similar reconstruction can be applied to GPU costs by estimating the per-operation GPU FLOPs and memory bandwidth utilization, which typically differ from CPU costs by implementation-specific constants but follow the same scaling laws.

The physical environment observations in our experiments are obtained from the analytical solution of the two-dimensional heat diffusion equation equation E.3 in free space. According to linear PDE theory, the temperature field evolution can be expressed through spatiotemporal convolution of the Green's function (heat kernel) with the heat source term:

$$T(\mathbf{x}, t) = T_\infty + \int_0^t \iint_{\mathbb{R}^2} G(\mathbf{x}, t; \boldsymbol{\xi}, \tau) Q(\boldsymbol{\xi}, \tau) \, d\boldsymbol{\xi} \, d\tau \tag{E.9}$$

where the heat kernel $G(\mathbf{x}, t; \boldsymbol{\xi}, \tau) = \frac{1}{4\pi\alpha(t-\tau)} \exp\left(-\frac{|\mathbf{x}-\boldsymbol{\xi}|^2}{4\alpha(t-\tau)}\right)$ describes the diffusion pattern of a unit point source in infinite space.

